# Utilitarian Algorithm Configuration

**Devon R. Graham**
Department of Computer Science
University of British Columbia
Vancouver, BC
drgraham@cs.cs.ubc.ca

**Kevin Leyton-Brown**
Department of Computer Science
University of British Columbia
Vancouver, BC
kevinlb@cs.ubc.ca

**Tim Roughgarden**
Columbia University & a16z crypto
New York, NY
tim.roughgarden@gmail.com

## Abstract

We present the first nontrivial procedure for configuring heuristic algorithms to maximize the utility provided to their end users while also offering theoretical guarantees about performance. Existing procedures seek configurations that minimize expected runtime. However, very recent theoretical work argues that expected runtime minimization fails to capture algorithm designers' preferences. Here we show that the utilitarian objective also confers significant algorithmic benefits. Intuitively, this is because mean runtime is dominated by extremely long runs even when they are incredibly rare; indeed, even when an algorithm never gives rise to such long runs, configuration procedures that provably minimize mean runtime must perform a huge number of experiments to demonstrate this fact. In contrast, utility is bounded and monotonically decreasing in runtime, allowing for meaningful empirical bounds on a configuration's performance. This paper builds on this idea to describe effective and theoretically sound configuration procedures. We prove upper bounds on the runtime of these procedures that are similar to theoretical lower bounds, while also demonstrating their performance empirically.

## 1 Introduction

Heuristic algorithms are surprisingly effective at solving hard computational problems. However, no single set of heuristics works well on all problems; the design choices must be tuned to a distribution of problem instances in the same way that the parameters of a machine learning model are tuned to a dataset. This black-box optimization problem is called *algorithm configuration*. Roughly speaking, the literature on algorithm configuration is divided into two camps. The older and larger camp designs heuristic procedures that aim to find good configurations as quickly as possible (Birattari et al., 2002; Hutter et al., 2009, 2011; Ansótegui et al., 2009; López-Ibáñez et al., 2016). A second, more recent line of work aims to provably identify approximately optimal configurations and to offer guarantees about the runtime required to do so (Kleinberg et al., 2017; Weisz et al., 2018; Kleinberg et al., 2019; Weisz et al., 2019, 2020). Virtually all of the work in both camps has focused on minimizing some version of capped average runtime, although one exception of which we are aware is Tornede et al. (2020), which explores objectives like higher-order polynomials of runtime that might better capture a user's level of risk-aversion.

Very recently, a theoretical argument has been made that minimizing expected runtime is inconsistent with a plausible set of axioms about algorithm designers' preferences (Graham et al., 2023). Runtime is an economic good and, like other economic goods, the value an individual assigns to it is not

necessarily proportional to the value of the good itself—nor should it be. We can characterize a set of bounded and monotonically decreasing utility functions that are implied by certain axioms. These are essentially the von Neumann–Morgenstern axioms (Von Neumann and Morgenstern, 1944) from classical decision theory, plus two novel runtime-specific axioms that assert a (weak) preference for faster runs over slower ones, and a (strict) preference for algorithm runs that complete over ones that time out. Preferences that follow these axioms are described by a utility function that is monotonically decreasing in runtime from 1 to 0. This utility function can incorporate factors like the benefit an end user attains from completing a run, the cost they pay for cloud computing resources, and any uncertainty they may have about when an answer will stop being useful to them. This utilitarian perspective opens a new and exciting direction for algorithm configuration.

While it is reassuring to know that optimizing utility instead of runtime is the right thing to do, it also offers significant algorithmic benefits. Because utilities are bounded and monotonically decreasing, exceedingly long runs contribute negligibly to estimates of an algorithm's expected utility. This means that there will always be some captime that allows us to accurately estimate true expected utility from capped samples. We can use tools from the multi-armed bandit literature for best arm identification, and the algorithms we describe will draw on the elimination and racing algorithms described in Mannor and Tsitsiklis (2004); Even-Dar et al. (2006) and elsewhere. In contrast, runtime-based configuration procedures like Structured Procrastination (Kleinberg et al., 2017), Structured Procrastination with Confidence (Kleinberg et al., 2019), LeapsAndBounds (Weisz et al., 2018), CapsAndRuns (Weisz et al., 2019), and ImpatientCapsAndRuns (Weisz et al., 2020) need to do many runs at large captimes in order to make theoretical guarantees. Furthermore, in order to be able to make any provable guarantees at all, they have to introduce an additional parameter specifying how much of the runtime CDF can be ignored; even if an algorithm has always finished very quickly on every instance seen so far, it could always take so long on the next instance that its expected runtime is arbitrarily large. In this setting, no optimality guarantees can be made without broadening the definition of optimality.

Other theoretically-motivated methods such as Gupta and Roughgarden (2017); Balcan et al. (2017, 2021) offer performance guarantees based on different measures of complexity and guarantee notions of PAC optimality akin to those we present here. These papers do not focus on runtime, instead studying traditional sample complexity, with each sample (i.e., algorithm run) contributing the same "sampling cost." In our algorithm configuration setting, the cost of a sample is the amount of (capped) runtime spent acquiring it, with longer captimes potentially leading to more expensive (but also more informative) samples. Because of this difference, these two lines of work propose quite different types of learning algorithms. However, Balcan et al. (2021) do assume the existence of a bounded utility function which measures algorithm performance, and others (Hoos and Stützle, 2004) have argued that utility functions of this form are better objectives to optimize than runtime when choosing algorithms. Some methods have been specifically designed to exploit the parallel nature of the algorithm configuration problem; AC-Band (Brandt et al., 2023) is a bandit-based procedure inspired by the Hyperband algorithm (Li et al., 2017) that runs multiple configurations simultaneously while ruling out poor-performing ones along the way and offering theoretical guarantees with respect to the set of configurations considered.

Inspired by the mantra of procrastination that has found success in previous work, and by the benefits that come with the use of utility functions, this paper presents a procedure we dub Utilitarian Procrastination (UP), so-named because it performs as many low-captime runs as possible before proceeding to higher-captime runs. We offer theoretical guarantees about its performance, showing that it will return a good configuration and proving that its worst-case upper bound is similar to the theoretical lower bound that any procedure must require. We also present experimental results showing how this procedure performs in practice and how it compares to a more naive baseline.

## 2   Setup

We assume there is a monotonically decreasing runtime utility function $u : \mathbb{R}_{\geq 0} \to [0, 1]$ with $u(0) = 1$ and $\lim_{t \to \infty} u(t) = 0$. The existence of $u$ follows from simple axioms (Graham et al., 2023). The value $u(t)$ describes the (expected) well-being of an individual who uses an algorithm to solve a problem instance (e.g., an integer program). During the configuration process, the goal will be to choose an algorithm that maximizes $u(t)$, in expectation over $t$.

Given a set of algorithms indexed by $i = 1, ..., n$, our goal is to find an approximately optimal algorithm using capped runtime samples. We assume we have access to a stream of input instances indexed by $j = 1, 2, ...$ drawn from some distribution $\mathcal{D}_J$. We will use $t_{ij}$ to denote the true uncapped runtime of algorithm $i$ on input $j$, and $t_{ij}(\kappa) = \min(t_{ij}, \kappa)$ to be the $\kappa$-capped runtime of $i$ on $j$. When we do a run, we observe $t_{ij}(\kappa)$, not $t_{ij}$. Of course these coincide for any run that completes. The instance distribution $\mathcal{D}_J$, along with any randomness of the algorithm or execution environment will together induce a runtime distribution for each algorithm $i$. We will use $\mathcal{D}_i$ to denote this runtime distribution, and $F_i$ to denote its CDF. For each algorithm $i$, the true uncapped expected utility is

$$U_i = \mathop{\mathbb{E}}_{t \sim D_i} \big[ u(t) \big].$$

The capped expected utility is

$$U_i(\kappa) = \mathop{\mathbb{E}}_{t \sim D_i} \big[ u\big( \min(t, \kappa) \big) \big].$$

And given any capped runtime samples $t_{i1}(\kappa), ..., t_{im}(\kappa)$, the capped empirical average utility is

$$\widehat{U}_{im}(\kappa) = \frac{1}{m} \sum_{j=1}^{m} u\big( t_{ij}(\kappa) \big).$$

**Definition 1** ($\epsilon$-optimal). *An algorithm $i^*$ is called $\epsilon$-optimal if $U_{i^*} \geq \max_i U_i - \epsilon$.*

Our goal will be to find $\epsilon$-optimal algorithms with probability at least $1 - \delta$, and to do so as quickly as possible. Our first lemma deterministically bounds an algorithm's uncapped expected utility in terms of its capped expected utility and CDF.

**Lemma 1.** *For any $i$ and $\kappa$ we deterministically have $U_i(\kappa) - u(\kappa)\big(1 - F_i(\kappa)\big) \leq U_i \leq U_i(\kappa)$.*

*Proof.* This follows from the law of total expectation and the fact that $u$ is non-increasing, and so $u(\kappa) \geq u(t)$ for all $t \geq \kappa$. □

Intuitively, Lemma 1 is true because the capped expected utility counts runs that cap as having just completed at the captime, when really they would have taken longer if given the chance. This makes the capped expected utility look more favourable than the uncapped expected utility. Our second lemma shows that if we do runs at captime $\kappa$, then we can accurately estimate the algorithm's runtime CDF at $\kappa$.

**Lemma 2.** *For any $i$, $m$, $\kappa$ and $\delta$, let $\widehat{F}_{im}(\kappa)$ be the fraction of the $m$ runs that $A$ completes within captime $\kappa$. Then $\widehat{F}_{im}(\kappa) - \sqrt{\frac{\ln(1/\delta)}{2m}} \leq F_i(\kappa) \leq \widehat{F}_{im}(\kappa) + \sqrt{\frac{\ln(1/\delta)}{2m}}$ with probability at least $1 - 2\delta$.*

*Proof.* For each $j$, let $X_j = 1$ if $t_{ij} < \kappa$ and $X_j = 0$ otherwise. The proof then follows from a straightforward application of Hoeffding's inequality. □

Lemma 2 simply says that if we take enough samples at captime $\kappa$, the fraction of those runs that complete will be close to the true likelihood of a run completing before $\kappa$. Our third lemma shows that expected capped utility can be estimated accurately using capped runtime samples.

**Lemma 3.** *For any $i$, $m$, $\kappa$ and $\delta$ we have $\widehat{U}_{im}(\kappa) - \big(1 - u(\kappa)\big)\sqrt{\frac{\ln(1/\delta)}{2m}} \leq U_i(\kappa) \leq \widehat{U}_{im}(\kappa) + \big(1 - u(\kappa)\big)\sqrt{\frac{\ln(1/\delta)}{2m}}$ with probability at least $1 - 2\delta$.*

*Proof.* Since $u\big(t_{ij}(\kappa)\big) \in [u(\kappa), 1]$ for all $j$, the proof follows immediately from Hoeffding's inequality. □

Lemma 3 simply says that if we take enough samples, then the empirical capped average utility will be close to the true expected capped utility. Together, these lemmas imply empirical confidence bounds on the true expected utility. Define the upper and lower confidence bounds

$$UCB_{im}(\kappa) = \widehat{U}_{im}(\kappa) + \big(1 - u(\kappa)\big)\sqrt{\frac{\ln(4n/\delta)}{2m}}$$

$$LCB_{im}(\kappa) = \widehat{U}_{im}(\kappa) - \sqrt{\frac{\ln(4n/\delta)}{2m}} - u(\kappa)(1 - \widehat{F}_{im}(\kappa)).$$

The next lemma shows that these are good confidence bounds if both the captime $\kappa$ and the number of samples $m$ are sufficiently large. With probability at least $1 - \delta$, each algorithm's true expected utility will fall within these confidence bounds, and the width of each confidence range will not be too large.

**Lemma 4.** *If we do $m$ runs of each algorithm at captime $\kappa$, then with probability at least $1 - \delta$ we will have*

$$LCB_{im} \leq U_i \leq UCB_{im}$$

*and*

$$UCB_{im} - LCB_{im} \leq 2\sqrt{\frac{\ln(4n/\delta)}{2m}} + u(\kappa)\big(1 - F_i(\kappa)\big)$$

*for all $i$ simultaneously.*

See Appendix A for a full proof. The idea is that the definition of the bounds together with Lemmas 2 and 3 mean the confidence bounds hold and are accurate.

We can see in Lemma 4 that the error in our estimate of an algorithm's expected utility comes from two sources: sampling and capping. The term $2\sqrt{\frac{\ln(4n/\delta)}{2m}}$ represents the error due to sampling, while the term $u(\kappa)\big(1 - F_i(\kappa)\big)$ represents the error due to capping. To make good guarantees, configuration procedures will need to ensure that both of these terms are sufficiently small.

# 3 Configuration Procedures

We first describe two hypothetical procedures that give us lower bounds on the number of samples and the captime that any configuration procedure will need to use. The bound on the number of samples (Section 3.1) is a classic result. In Section 3.2, we use a novel "prover-skeptic" argument to show the lower bound on captime. We then describe in Section 3.3 a simple usable procedure that returns an $\epsilon$-optimal algorithm with probability at least $1 - \delta$, but which suffers from two major drawbacks. First, it requires that we specify an accuracy parameter $\epsilon$ and a captime $\kappa$ as input ahead of time. Making poor choices for these parameters can have a large impact on total configuration time (see Section 4 for an illustration). Indeed, many choices of $\epsilon$, and $\kappa$ are mutually incompatible, giving rise to meaningless bounds. To avoid this, we hope to design procedures that are *anytime*: they continue to improve the accuracy guarantees they can make as they spend more time running. By taking gradually more and more samples, and by gradually taking them at higher and higher captimes, an anytime procedure continues to shrink the $\epsilon$ that it can guarantee, rather than trying to target a fixed $\epsilon$ it is given as input. Second, the naive procedure is not input-adaptive in any way. It does the same number of samples at the same captime for every algorithm. Both of these drawbacks are fixed by our UP procedure in Section 3.4. UP is an anytime procedure, meaning it requires neither an $\epsilon$ nor a $\kappa$ as input, but instead gradually reduces the $\epsilon$ it can guarantee by increasing both the number of samples and the captime.

## 3.1 Hypothetical Runtime Oracle Procedure

The unique characteristic of our setting is the fact that we observe capped rather than true runtime samples, and that the cost we pay for each sample is equal to the time we spend collecting it. If this were not the case, then our problem would already be solved. If we had some oracle that we could simply query for the true runtime of an algorithm run on a given instance, then all we would need to do is collect sufficiently many samples from each algorithm. In this case, the optimal procedure simply takes an increasing number of samples of each algorithm and rules out each suboptimal one was soon as it can. The Runtime Oracle Procedure (Algorithm 1) is an instantiation of the Successive Elimination algorithm of Even-Dar et al. (2006) (their Algorithm 3). With probability at least $1 - \delta$ it will eventually eliminate all algorithms except the optimal one.

**Theorem 1.** *With probability at least $1 - \delta$ the Runtime Oracle procedure will eventually return the optimal algorithm, and it will return an $\epsilon$-optimal algorithm if it is run until $m$ is large enough that $2\sqrt{\frac{\ln(4nm^2/\delta)}{2m}} \leq \epsilon$. At that point it will have taken $m$ uncapped samples of each $\epsilon$-optimal algorithm,*

---

**Algorithm 1** Runtime Oracle Procedure

---

**Inputs:** Algorithms $i = 1, ..., n$; stream of instance $j = 1, 2, ...$; utility function $u$; parameter $\delta$; runtime oracle $\mathcal{R}$.

$I \leftarrow \{1, ..., n\}$           ▷ candidate algorithms
**for** $m = 1, 2, 3, ....$ **do**
     **for** $i \in I$ **do**
         $t_{im} \leftarrow$ obtain $i$'s runtime from $\mathcal{R}$ on $m$-th instance
         $\widehat{U}_{im} \leftarrow \frac{1}{m} \sum_{j=1}^{m} u(t_{ij})$
     **end for**
     $i^* \leftarrow \arg\max_{i \in I} \widehat{U}_{im}$
     $\alpha_m \leftarrow \sqrt{\frac{\ln(4nm^2/\delta)}{2m}}$
     **for** $i \neq i^*$ **do**
         **if** $\widehat{U}_{im} < \widehat{U}_{i^*m} - 2\alpha_m$ **then**          ▷ $i$ is suboptimal
             $I \leftarrow I \setminus \{i\}$          ▷ remove $i$
         **end if**
     **end for**
     **if** $|I| = 1$ **or** execution is interrupted **then**
         **return** $i^*$
     **end if**
**end for**

---

*and $m_i \leq m$ uncapped samples of each $\epsilon$-suboptimal algorithm $i$, where $2\sqrt{\frac{\ln(4nm_i^2/\delta)}{2m_i}} \leq \Delta_i$, and where $\Delta_i$ is the difference between the expected utility of an optimal algorithm and of algorithm $i$.*

See Even-Dar et al. (2006), Theorem 8 and Remark 9 for proofs and details. The Runtime Oracle procedure has two desirable qualities that we would like to preserve in the configuration procedures we design. Firstly, we do not need to supply the parameter $\epsilon$ ahead of time. Instead, the procedure maintains an internal $\epsilon$ that it can guarantee, continually shrinking this toward 0. Secondly, the number of samples needed to eliminate $\epsilon$-suboptimal algorithms grows with the square of the suboptimality gap $\Delta_i = \max_{i'} U_{i'} - U_i$ rather than the square of $\epsilon$. Theorem 1 gives an input-dependent guarantee for the Runtime Oracle procedure that we would like to preserve in the usable procedures we design. If some algorithm $i$ is very suboptimal, with $\Delta_i \gg \epsilon$, then we will be able to eliminate it with many fewer samples than an algorithm that is almost $\epsilon$-optimal.

The Runtime Oracle procedure gives us a baseline against which to compare. Its sample complexity guarantee is essentially optimal in the following data-dependent sense.

**Theorem 2** (Theorem 5 of Mannor and Tsitsiklis (2004)). *There exists some set of input distributions for which every oracle procedure that returns an $\epsilon$-optimal algorithm with probability at least $1 - \delta$ must take $\Omega\left(\frac{n - |N(\epsilon)|}{\epsilon^2} \log \frac{1}{\delta} + \sum_{i \in N(\epsilon)} \frac{1}{\Delta_i^2} \log \frac{1}{\delta}\right)$ samples, where $N(\epsilon)$ is the set of $\epsilon$-suboptimal algorithms.*

### 3.2 Hypothetical Captime Verification Procedure

We now consider a lower bound on the captime we will be required to use, even in a world where we do not need to take samples. We imagine there are a *prover* and a *skeptic*. The prover has access to the runtime CDF of each algorithm $i$. The skeptic will get to see each runtime CDF only up to some $\kappa_i$ that the prover will choose. The prover then recommends an algorithm $i^*$ and the skeptic should be convinced that $i^*$ is $\epsilon$-optimal, based only on the truncated CDFs. As the $\kappa_i$'s tend to infinity, the skeptic will see more and more of the true CDFs and therefore eventually be convinced of the prover's claim (assuming it is true). The goal is to convince the skeptic that $i^*$ is $\epsilon$-optimal using $\kappa_i$'s that are as small as possible. The skeptic will be able to compute the following deterministic bounds

---

**Algorithm 2** Captime Verification Procedure

---

**Inputs:** Algorithms $i = 1, ..., n$; stream of instances $j = 1, 2, ...$; utility function $u$; parameter $\epsilon$; runtime CDFs $F_1, ..., F_n$.

**for** $i = 1, ..., n$ **do**
    $U_i \leftarrow \int_0^\infty u(t)dF_i(t)$                                      ▷ expected utility
**end for**
$i^{opt} \leftarrow \arg\max_i U_i$                                          ▷ optimal algorithm
**for** $i = 1, ..., n$ **do**
    $\Delta_i \leftarrow U_{i^{opt}} - U_i$
    $\kappa_i \leftarrow \inf \left\{ \kappa \ : \ u(\kappa)\big(1 - F_i(\kappa)\big) \leq \Delta_i + \frac{\epsilon}{2} \right\}$
**end for**
$i^* = \arg\max_i LB_i$
**if** $LB_{i^*} \geq UB_i - \epsilon$ **for** $i = 1, ..., n$ **then**                  ▷ skeptic's check
    **return** $i^*$
**else**
    **return** "failed"
**end if**

---

on an algorithm's expected utility

$$UB_i = \int_0^{\kappa_i} u(t)dF_i(t) + u(\kappa_i)\big(1 - F_i(\kappa_i)\big)$$

$$LB_i = \int_0^{\kappa_i} u(t)dF_i(t).$$

The skeptic knows that $LB_i \leq U_i \leq UB_i$, but also that the runtime distributions could be such as to make $U_i$ fall anywhere in this range. So whatever algorithm $i^*$ the prover returns to the skeptic, the skeptic will be convinced that $i^*$ is $\epsilon$-optimal if, and only if, they observe $LB_{i^*} \geq UB_i - \epsilon$ for all $i \neq i^*$. The following lemma justifies this condition.

**Lemma 5.** *If $LB_{i^*} \geq UB_i - \epsilon$ for all $i \neq i^*$, then $i^*$ is $\epsilon$-optimal. If there exists an $i \neq i^*$ with $LB_{i^*} < UB_i - \epsilon$, then regardless of the values of the CDFs $F_i$ up to the captimes $\kappa_i$, there are some input distributions for which $i^*$ is not $\epsilon$-optimal.*

The proof works by constructing counter-example CDFs, but takes the CDFs as given up $\kappa_i$, so does not rely on specifying any information available to the skeptic. No matter what the CDF of each $i$ looks like up to $\kappa_i$, there will be some inputs with $LB_{i^*} < UB_i - \epsilon$ for which $i^*$ is not $\epsilon$-optimal.

**Lemma 6.** *The skeptic will be convinced that both $i^{opt}$ and $i^* = \arg\max_i LB_i$ are $\epsilon$-optimal if the captimes $\kappa_i$ are large enough that $u(\kappa_i)\big(1 - F_i(\kappa_i)\big) \leq \Delta_i + \frac{\epsilon}{2}$ for all algorithms $i$.*

Algorithm 2 describes the prover-skeptic interaction. The next lemma shows that using these captimes is essentially optimal. It gives us a baseline minimum captime against which to compare.

**Lemma 7.** *There are some inputs for which any verification procedure must use captimes $\kappa_i$ large enough that $u(\kappa_i)\big(1 - F_i(\kappa_i)\big) \leq \Delta_i + \epsilon$ for all algorithms $i$, or the skeptic will not be convinced that the returned algorithm is $\epsilon$-optimal.*

The proof simply constructs a counterexample for which doing runs at a smaller captime will lead any configuration procedure to make a wrong conclusion.

## 3.3 Naive Procedure

We now turn to the setting we are actually interested in, where runs cost time and long ones must eventually be terminated. The simplest thing we could do is just pick a captime (or perhaps we somehow "know" the right captime) and do the necessary number of runs. With a fixed captime, the scenario is not much different from the runtime oracle scenario above. The main difference is that the "resolution" at which we can understand an algorithm's expected utility will be limited by the size of the captime we use to take our samples. We will never know what the runtime CDF looks like beyond the captime, no matter how many samples we take. Given any $\kappa$, the confidence bounds described in

---
**Algorithm 3** Naive Procedure
---
**Inputs:** Algorithms $i = 1, ..., n$; stream of instances $j = 1, 2, ...$; utility function $u$; parameters $\epsilon, \delta$; captime $\kappa$ satisfying $u(\kappa) < \epsilon$.

$m \leftarrow \left\lceil \frac{2 \ln(2n/\delta)}{(\epsilon - u(\kappa))^2} \right\rceil$

**for** $i = 1, ..., n$ **do**
    $t_{i1}(\kappa), ..., t_{im}(\kappa) \leftarrow$ run $i$ on $m$ instances using captime $\kappa$
    $\widehat{U}_{im}(\kappa) \leftarrow \frac{1}{m} \sum_{j=1}^{m} u(t_{ij}(\kappa))$                     ▷ empirical average utility
**end for**
$i^* \leftarrow \arg\max_i \widehat{U}_{im}(\kappa)$
**return** $i^*$

---

Section 2 tell us we could simply choose the smallest $m$ satisfying $2\sqrt{\frac{\ln(2n/\delta)}{2m}} + u(\kappa) \le \epsilon$, do $m$ runs of each algorithm at captime $\kappa$, then return the one with largest empirical average utility. We can think of $2\sqrt{\frac{\ln(2n/\delta)}{2m}}$ as the error due to sampling, and $u(\kappa)$ as the error due to capping. This is the Naive procedure.

**Theorem 3.** *With probability at least $1 - \delta$ the Naive procedure returns an $\epsilon$-optimal algorithm. It takes enough $\kappa$-capped runtime samples of each algorithm to ensure that $2\sqrt{\frac{\ln(2n/\delta)}{2m}} + u(\kappa) \le \epsilon$.*

The proof of this and of subsequent theorems are deferred to Appendix A. Each essentially follows from the fact that the upper and lower confidence bounds are specified to satisfy Hoeffding's inequality and the union bound. Comparing Theorem 3 to Theorem 1 we can see the reason for the increased number of samples required when we observe capped instead of uncapped runs: if $u(\kappa)$ is relatively large, it will take a much larger $m$ to shrink the term $2\sqrt{\frac{\ln(2n/\delta)}{2m}}$ enough to satisfy the inequality in Theorem 3.

The Naive procedure has some undesirable qualities. Choosing the right $\kappa$ may not be easy and in Section 4 we will show what a difference this choice can make for the total runtime of the Naive procedure. We are also required to specify an $\epsilon$ beforehand; it may hard to know the "right" $\epsilon$, and the best $m$ and $\kappa$ for one $\epsilon$ may be quite different from the best for a slightly different $\epsilon$. What's more, this procedure ignores information about an algorithm's runtime distribution that could be learned by observing runs along the way; it essentially assumes that every algorithm always times out at the given captime. As a result, it takes more samples of $\epsilon$-suboptimal algorithms than is necessary. The UP procedure corrects these defects.

### 3.4 Utilitarian Procrastination

Our Utilitarian Procrastination (UP) procedure starts by doing runs at the smallest captime possible, trying to rule out whatever configurations we can, and only starts doing runs at a larger captime when necessary.

**Theorem 4.** *With probability at least $1 - \delta$ UP eventually returns the optimal algorithm and it returns an $\epsilon$-optimal algorithm if it is run until $m$ is large enough that $2\sqrt{\frac{\ln(11nm^2(\log \kappa_{i^{opt}} + 1)^2/\delta)}{2m}} + u(\kappa_{i^{opt}})\big(1 - F_{i^{opt}}(\kappa_{i^{opt}})\big) \le \epsilon$. For any suboptimal $i$, if $m, \kappa_i, \kappa_{i^{opt}}$ are ever large enough that $2\sqrt{\frac{\ln(11nm^2(\log \kappa_i + 1)^2/\delta)}{2m}} + 2\sqrt{\frac{\ln(11nm^2(\log \kappa_{i^{opt}} + 1)^2/\delta)}{2m}} + u(\kappa_i)\big(1 - F_i(\kappa_i)\big) + u(\kappa_{i^{opt}})\big(1 - F_{i^{opt}}(\kappa_{i^{opt}})\big) \le \Delta_i$ then $i$ will be eliminated.*

In Theorem 4 we can clearly see the analog of both Theorem 1 and Theorem 3. UP is input dependent in two senses. The condition for ruling out a suboptimal $i$ depends on $\Delta_i$ and also on the runtime CDFs $F_i$ and $F_{i^{opt}}$. Just as in the pure sample complexity case of Theorem 1, if $i$ is very suboptimal and $\Delta_i$ is large, it will be easier to satisfy the condition in Theorem 4 and thus rule $i$ out. Additionally, the condition will be easier to satisfy if the inputs are such that either $i$ or $i^{opt}$ is able to complete a lot of runs at their respective captimes, so that their CDFs are close to 1. Theorem 4 also implies the following theorem and corollary, which together constitute our main theoretical result.

---

**Algorithm 4** Utilitarian Procrastination

---

**Inputs:** Algorithms $i = 1, ..., n$; stream of instances $j = 1, 2, ...$; utility function $u$; parameter $\delta$.
$I \leftarrow \{1, ..., n\}$ ▷ candidate algorithms
$\kappa_i \leftarrow 1$ **for all** $i \in I$
**for** $m = 1, 2, 3, ...$ **do**
    **for** $i \in I$ **do**
        $t_{i1}(\kappa_i), ..., t_{im}(\kappa_i) \leftarrow$ run $i$ on $m$ instances using captime $\kappa_i$
        $\widehat{F}_{im}(\kappa_i) \leftarrow \frac{|\{j : t_{ij}(\kappa_i) < \kappa_i\}|}{m}$ ▷ fraction of runs that completed
        $\widehat{U}_{im}(\kappa_i) \leftarrow \frac{1}{m} \sum_{j=1}^{m} u(t_{ij}(\kappa_i))$ ▷ empirical average utility
        $\alpha_{im} \leftarrow \sqrt{\frac{\ln(11nm^2(\log \kappa_i + 1)^2/\delta)}{2m}}$
        $UCB_{im} \leftarrow \widehat{U}_{im}(\kappa_i) + (1 - u(\kappa_i))\alpha_{im}$
        $LCB_{im} \leftarrow \widehat{U}_{im}(\kappa_i) - \alpha_{im} - u(\kappa_i)(1 - \widehat{F}_{im}(\kappa_i))$
    **end for**
    $i^* \leftarrow \arg\max_{i \in I} LCB_{im}$
    **for** $i \in I$ **do**
        **if** $UCB_{im} < LCB_{i^*m}$ **then** ▷ $i$ is suboptimal
            $I \leftarrow I \setminus \{i\}$ ▷ remove $i$
        **end if**
        **if** $2\alpha_{im} \leq u(\kappa_i)(1 - \widehat{F}_{im}(\kappa_i))$ **then** ▷ captime doubling condition
            $\kappa_i \leftarrow 2\kappa_i$
        **end if**
    **end for**
    **if** $|I| = 1$ **or** execution is interrupted **then**
        **return** $i^*$
    **end if**
**end for**

---

**Theorem 5.** *For any $m$, let $\epsilon = 3\sqrt{\frac{\ln(11nm^4/\delta)}{2m}}$. Then with probability at least $1 - \delta$, UP returns an $\epsilon$-optimal algorithm once it takes $m$ samples. At that point, if $\kappa_i^*$ is the largest captime algorithm $i$ has been run with then $\kappa_i^* \leq 2\inf\{\kappa : u(\kappa)(1 - F_i(\kappa)) < \frac{\epsilon}{3\sqrt{2}}\}$.*

The proof follows from Theorem 4 and from UP's specific choice of captime doubling condition. We note that there is room for improvement, but that this captime bound is comparable to the worst-case lower bound captime needed by any configuration procedure as presented in Section 3.2. The smallest captime that satisfies $u(\kappa)(1 - F_i(\kappa)) < \frac{\epsilon}{3\sqrt{2}}$ needs to be larger than the smallest captime that satisfies $u(\kappa)(1 - F_i(\kappa)) < \Delta_i + \epsilon$, which is the best we can hope to do.

**Corollary 1.** *With probability at least $1 - \delta$, UP returns an $\epsilon$-optimal algorithm after taking a number of samples that at most a logarithmic factor more than is optimally required by any procedure, and at a captime that is a constant larger than $\inf\{\kappa : u(\kappa)(1 - F_i(\kappa)) < \frac{\epsilon}{3\sqrt{2}}\}$.*

*Proof.* The number of samples follows immediately from the definition of $\epsilon$ in Theorem 5. Because the captimes are always doubled, the total time spent by UP on any single instance is at most a constant times larger than the time spent running that instance the final time it was run. Since the captime at that point was at most $\kappa_i^*$, the total time spent running any instance is at most a constant times $\inf\{\kappa : u(\kappa)(1 - F_i(\kappa)) < \frac{\epsilon}{3\sqrt{2}}\}$. □

## 4 Experiments

We now illustrate the runtime costs of utilitarian algorithm configuration and the impacts of the adaptive improvements offered by UP over Naive. We would have liked to go further, and in particular to offer comparisons against baselines other than Naive. However, we saw no straightforward way of doing so: there is simply no previous work on offering algorithm configuration in the utility-maximizing setting. Our paper focuses on algorithm configuration with theoretical guarantees. There

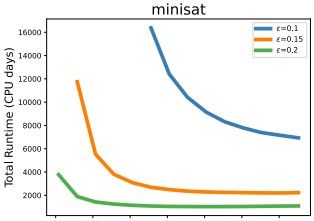 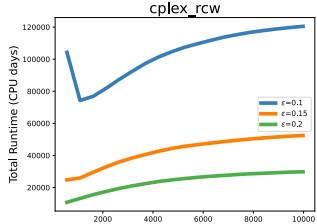 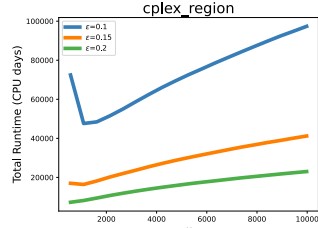

Figure 1: Runtime of the Naive procedure for different values of input captime $\kappa$ on three datasets using a log-Laplace utility function. Choosing a bad captime can have a large effect on total configuration time, especially for smaller $\epsilon$ values.

do exist many methods with guarantees in the runtime minimizing setting; we could have run existing procedures like Structured Procrastination, LeapsAndBounds, or CapsAndRuns and then evaluated them in terms of a utilitarian objective. However, such an apples-to-oranges comparison would likely have yielded very poor performance and regardless would have dispensed with the guarantees that motivate these methods. Second, we could have compared to heuristic methods like SMAC, ParamILS and GGA. Again, however, they optimize a different objective function. It is possible to imagine modifying one of these algorithms to optimize utility, but doing so would require fundamental algorithmic changes (e.g., to their so-called adaptive capping heuristics) and nontrivial software engineering effort. Even if we could have made such changes in a non-controversial way, comparing heuristic methods to methods offering guarantees would again be an apples-to-oranges comparison. Perhaps for these reasons, we note that no paper of which we are aware has yet compared any heuristic algorithm configuration method to any offering theoretical guarantees. We do intend to pursue such an investigation ourselves in future work, but anticipate that a careful study of this question will require an entire research paper.

**Experimental Setup.** We leverage three datasets from Weisz et al. (2020). The first is a set of runtimes for the minisat SAT solver on data generated by the CNFuzzdd instance generator. The others are sets of runtimes for the CPLEX integer program solver on the combinatorial auction winner determination instances (regions) and on woodpecker conservation problems (rcw); see Appendix D of Weisz et al. (2020) for details. We only used the first seed for the CPLEX datasets.[1]

**Choosing the Right Captime Matters.** In Fig. 1 we plot the total configuration time of the Naive procedure with different input captimes $\kappa$ on three datasets from Weisz et al. (2020) using a log-Laplace utility function from Graham et al. (2023): $u(t) = u_{LL}(t; 60, 1) = 1 - \frac{1}{2}\frac{t}{60}$ if $t \leq 60$ and $u(t) = \frac{1}{2}\frac{60}{t}$ otherwise. We used $\epsilon$ values of 0.1, 0.15 and 0.2 and set $\delta = 0.1$. The first observation we make is that choosing a bad captime can have a large effect on total configuration time, especially for smaller $\epsilon$ values. The second observation is that, for any fixed $\kappa$, different values of $\epsilon$ can also have huge effects on total runtime of the configuration procedure. Both of these points help to emphasize the need for a procedure like UP that starts out with a small $\kappa$, only increasing it as needed, and refines the $\epsilon$ it can guarantee "on the fly", based on the runs it has observed.

**Anytime Speedups Matter.** In Fig. 2, we plot the total configuration time of UP and Naive as a function of $\epsilon$. We used the same log-Laplace utility function as above and set $\delta = 0.1$ (within reasonable ranges, the value of $\delta$ has relatively little effect on total runtime). UP can drastically outperform Naive, especially for smaller values of $\epsilon$.

Unsurprisingly, being anytime in $\kappa$ helps most when we do not know how to provide a good $\kappa$ as input beforehand. If we somehow guess or know the right captime to use, then it can be hard to beat Naive. But if we use the wrong captime, UP can be much faster than Naive. In Fig. 3 we use a uniform utility function from Graham et al. (2023), with $u(t) = u_{unif}(t; 60) = 1 - \frac{t}{60}$ for $t < 60$ and 0 otherwise. In the top row, we have used a captime of $\kappa = 60$ for Naive, which is appropriate for this utility function since it is linear up to that point and then 0 thereafter. In the bottom row, we have set the captime poorly at $\kappa = 600$, meaning that we are not terminating some runs even though

---

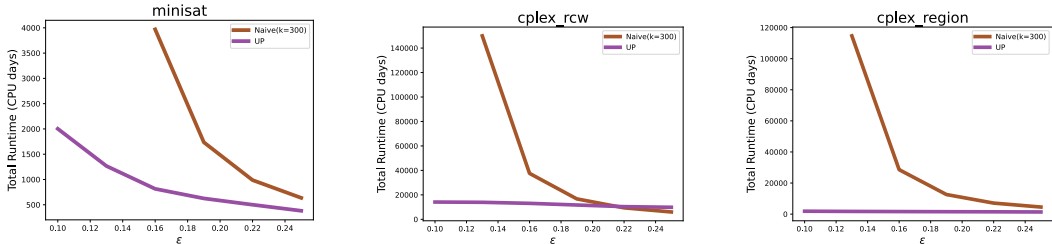

Figure 2: Runtime of the different configuration procedures for different values of $\epsilon$ on three different datasets using a log-Laplace utility function. UP easily outperforms Naive.

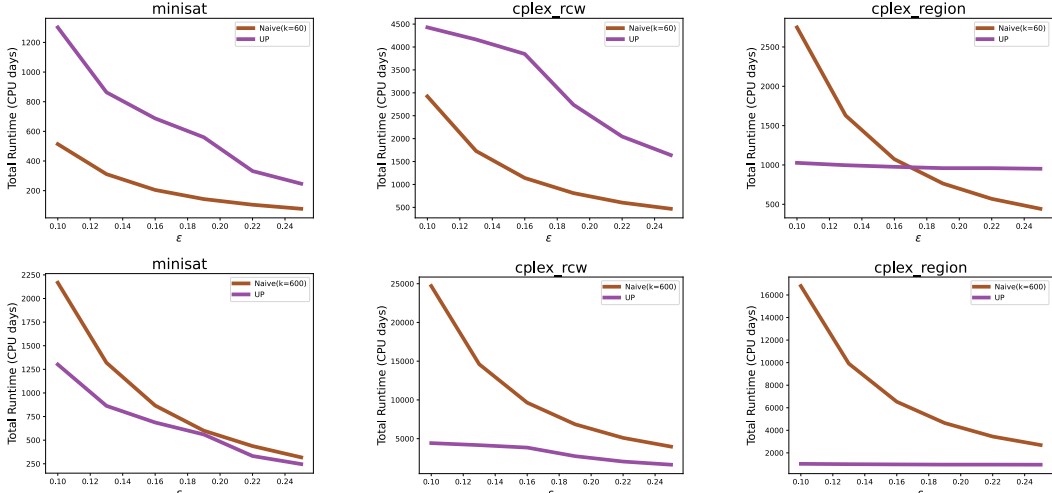

Figure 3: Runtime of the different configuration procedures for different values of $\epsilon$ on three different datasets using a uniform utility function. The top row shows a scenario where the Naive procedure has been given an appropriate captime, the bottom row shows a scenario where it has not.

they are so long they give us $0$ utility. We can see that if we get the captime right, then Naive will perform quite well. But if we get it wrong, Naive can drastically underperform compared to UP.

## 5    Conclusion

We have presented Utilitarian Procrastination (UP), a utility-based approach to automated algorithm configuration. This is the first procedure that we know of to incorporate utility functions into the algorithm configuration framework. UP is anytime, meaning it requires minimal input from the user and continues to refine its guarantees as it is run. In simple experiments, we show that by freeing the user from having to provide either an accuracy parameter $\epsilon$ or a captime $\kappa$ UP can help avoid the excessively long runtimes that come when these inputs are chosen poorly. Of course we would have liked to have shown that UP never uses a captime larger than $\kappa_i^* \leq 2\inf \left\{\kappa \ : \ u(\kappa)\big(1 - F_i(\kappa)\big) < \Delta_i + \frac{\epsilon}{2}\right\}$, matching the hypothetical procedure. The bound we do show is a result of the condition UP checks when deciding whether to double the the captime $\kappa_i$. UP uses a simple condition that balances the error due to sampling with the error due to capping. Other, more intelligent choices may be possible, but improving this will have to wait for future work.

# 6  Acknowledgements

Graham and Leyton-Brown were funded by an NSERC Discovery Grant, a DND/NSERC Discovery Grant Supplement, a CIFAR Canada AI Research Chair (Alberta Machine Intelligence Institute), a Compute Canada RAC Allocation, and DARPA award FA8750-19-2-0222, CFDA #12.910 (Air Force Research Laboratory). Roughgarden's research at Columbia University is supported in part by NSF awards CCF-2006737 and CNS-2212745.

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
