*Proof.* Define the good events:

$$\mathcal{C}_i = \left\{ \widehat{F}_i(\kappa) - \sqrt{\frac{\ln(4n/\delta)}{2m}} \leq F_i(\kappa) \leq \widehat{F}_i(\kappa) + \sqrt{\frac{\ln(4n/\delta)}{2m}} \right\}$$

$$\mathcal{B}_i = \left\{ \widehat{U}_i - \big(1 - u(\kappa)\big)\sqrt{\frac{\ln(4n/\delta)}{2m}} \leq U_i^{(\kappa)} \leq \widehat{U}_i + \big(1 - u(\kappa)\big)\sqrt{\frac{\ln(4n/\delta)}{2m}} \right\}.$$

Then by Lemmas 2 and 3 the good events hold simultaneously for all $i$ simultaneously with probability at least $1 - \delta$.

If the good events hold, we will have

$$
\begin{aligned}
LCB_i = \widehat{U}_i - u(\kappa)(1 - \widehat{F}_i(\kappa)) - \sqrt{\frac{\ln(4n/\delta)}{2m}} && \text{(definition)} \\
\leq \widehat{U}_i - u(\kappa)\left(1 - F_i(\kappa) - \sqrt{\frac{\ln(4n/\delta)}{2m}}\right) - \sqrt{\frac{\ln(4n/\delta)}{2m}} && \text{(good events)} \\
\leq U_i^{(\kappa)} - u(\kappa)\big(1 - F_i(\kappa)\big) && \text{(good events)} \\
\leq U_i && \text{(Lemma 1)} \\
\leq U_i^{(\kappa)} && \text{(Lemma 1)} \\
\leq \widehat{U}_i + \big(1 - u(\kappa)\big)\sqrt{\frac{\ln(4n/\delta)}{2m}} && \text{(good events)} \\
= UCB_i && \text{(definition).}
\end{aligned}
$$

And also

$$
\begin{aligned}
UCB_i - LCB_i = u(\kappa)\big(1 - \widehat{F}_{i,\ell}\big) + \big(2 - u(\kappa)\big)\sqrt{\frac{\ln(4n/\delta)}{2m}} && \text{(definition)} \\
\leq u(\kappa)\big(1 - F_i(\kappa)\big) + 2\sqrt{\frac{\ln(4n/\delta)}{2m}} && \text{(good events).}
\end{aligned}
$$

$\square$

**Lemma 5:** If $LB_{i^*} \geq UB_i - \epsilon$ for all $i \neq i^*$, then $i^*$ is $\epsilon$-optimal. If there exists an $i$ with $LB_{i^*} < UB_i - \epsilon$, then regardless of the values of the CDFs $F_i$ up to the captimes $\kappa_i$, there are some input distributions for which $i^*$ is not $\epsilon$-optimal.

*Proof.* If the skeptic observes $LB_{i^*} \geq UB_i - \epsilon$ for all $i \neq i^*$, then they will know that either $i^* = i^{opt}$ or that

$$
\begin{aligned}
U_{i^*} \geq LB_{i^*} && \text{(Lemma 1)} \\
\geq UB_{i^{opt}} - \epsilon && \text{(observed)} \\
\geq U_{i^{opt}} - \epsilon && \text{(Lemma 1),}
\end{aligned}
$$

so they will be convinced that $i^*$ is $\epsilon$-optimal.

Let $u^{-1}(x) = \inf\{t \ : \ u(t) \leq x\}$ be the smallest runtime for which the utility falls below $x$. Suppose there is some $i \neq i^*$ for which $UB_i > LB_{i^*} + \epsilon$. Then it may be the case that $i^*$ is not actually $\epsilon$-optimal: Let $\alpha = 2(UB_i - LB_{i^*})$ and observe that $\alpha > 2\epsilon$. Suppose the distribution

of $i$ is such that $F_i(t) = F_i(\kappa_i)$ for all $t \in [\kappa_i, A)$ and $F_i(t) = 1$ for $t \geq A$, where $A = 2\kappa_i$ if $u(\kappa_i)\big(1 - F_i(\kappa_i)\big) \leq \frac{\alpha}{4}$ and $A = u^{-1}\Big(u(\kappa_i) - \frac{\alpha}{4\big(1 - F_i(\kappa_i)\big)}\Big)$ otherwise. Then we will have

$$\int_{\kappa_i}^\infty u(t)dF_i(t) = u(A)\big(1 - F_i(\kappa_i)\big)$$

$$\geq u(\kappa_i)\big(1 - F_i(\kappa_i)\big) - \frac{\alpha}{4}.$$

Suppose the distribution of $i^*$ is such that $F_{i^*}(t) = F_{i^*}(\kappa_{i^*})$ for all $t \in [\kappa_{i^*}, B)$ and $F_{i^*}(t) = 1$ for $t \geq B$, where $B = 2\kappa_{i^*}$ if $u(\kappa_{i^*})\big(1 - F_{i^*}(\kappa_{i^*})\big) \leq \frac{\alpha}{4}$ and $B = u^{-1}\Big(\frac{\alpha}{4\big(1 - F_{i^*}(\kappa_{i^*})\big)}\Big)$ otherwise. Then we will have

$$\int_{\kappa_{i^*}}^\infty u(t)dF_{i^*}(t) = u(B)\big(1 - F_{i^*}(\kappa_{i^*})\big)$$

$$\leq \frac{\alpha}{4}.$$

Thus, with these distributions we will have

$$
\begin{aligned}
U_i &= UB_i - u(\kappa_i)\big(1 - F_i(\kappa_i)\big) + \int_{\kappa_i}^\infty u(t)dF_i(t) && \text{(definition of } UB_i) \\
&\geq UB_i - \frac{\alpha}{4} && \text{(choice of } F_i) \\
&= LB_{i^*} + \frac{\alpha}{4} && \text{(assumption)} \\
&= U_{i^*} - \int_{\kappa_{i^*}}^\infty u(t)dF_{i^*}(t) + \frac{\alpha}{4} && \text{(definition of } LB_{i^*}) \\
&\geq U_{i^*} && \text{(choice of } F_{i^*}) \\
&> U_{i^*} - \epsilon && \text{(negative term)}
\end{aligned}
$$

so $i^*$ is not $\epsilon$-optimal. $\qquad\square$

**Lemma 6:** The skeptic will be convinced that both $i^{opt}$ and $i^* = \arg\max_i LB_i$ are $\epsilon$-optimal if the captimes $\kappa_i$ are large enough that $u(\kappa_i)\big(1 - F_i(\kappa_i)\big) \leq \Delta_i + \frac{\epsilon}{2}$ for all algorithms $i$.

*Proof.* Suppose that $u(\kappa_i)\big(1 - F_i(\kappa_i)\big) \leq \Delta_i + \frac{\epsilon}{2}$ for all $i$. Then we will have

$$
\begin{aligned}
UB_i &= LB_i + u(\kappa_i)\big(1 - F_i(\kappa_i)\big) && \text{(definition of } UB_i) \\
&\leq LB_i + \Delta_i + \frac{\epsilon}{2} && (\kappa_i \text{ large enough)} \\
&\leq U_i + \Delta_i + \frac{\epsilon}{2} && \text{(Lemma 1)} \\
&= U_{i^{opt}} + \frac{\epsilon}{2} && \text{(definition of } \Delta_i) \\
&\leq UB_{i^{opt}} + \frac{\epsilon}{2} && \text{(Lemma 1} \\
&= LB_{i^{opt}} + u(\kappa_{i^{opt}})\big(1 - F_{i^{opt}}(\kappa_{i^{opt}})\big) + \frac{\epsilon}{2} && \text{(definition of } UB_{i^{opt}}) \\
&\leq LB_{i^{opt}} + \epsilon && (\kappa_i \text{ large enough)} \\
&\leq LB_{i^*} + \epsilon && \text{(choice of } i^*)
\end{aligned}
$$

so the skeptic can apply Lemma 5 and be convinced that $i^{opt}$ and $i^*$ are $\epsilon$-optimal. $\qquad\square$

**Lemma 7:** There are some inputs for which any verification procedure must use captimes $\kappa_i$ large enough that $u(\kappa_i)\big(1 - F_i(\kappa_i)\big) \leq \Delta_i + \epsilon$ for all algorithms $i$, or the skeptic will not be convinced that the returned algorithm is $\epsilon$-optimal.

*Proof.* Suppose $u(\kappa_i)\big(1 - F_i(\kappa_i)\big) > \Delta_i + \epsilon$ for some $i$. Let $u^{-1}(x) = \inf\{t \ : \ u(t) \le x\}$ be the smallest runtime for which the utility falls below $x$. Suppose the distribution of $i$ is such that $F_i(t) = F_i(\kappa_i)$ for $t \in (\kappa_i, A)$ where $A = u^{-1}\Big(\frac{1}{2}\Big(u(\kappa_i) - \frac{\Delta_i + \epsilon}{1 - F_i(\kappa_i)}\Big)\Big)$ and $F_i(t) = 1$ for $t \ge A$. Note that $u(A)\big(1 - F_i(\kappa_i)\big) < u(\kappa_i)\big(1 - F_i(\kappa_i)\big) - \Delta_i - \epsilon$. Let $i^*$ be the returned algorithm. We have

$$
\begin{aligned}
UB_i &= U_i + u(\kappa_i)\big(1 - F_i(\kappa_i)\big) - \int_{\kappa_i}^{\infty} u(t)dF_i(t) &&\text{(definition of } UB_i) \\
&= U_i + u(\kappa_i)\big(1 - F_i(\kappa_i)\big) - u(A)\big(1 - F_i(\kappa_i)\big) &&\text{(choice of } F_i) \\
&= U_{i^{opt}} - \Delta_i + u(\kappa_i)\big(1 - F_i(\kappa_i)\big) - u(A)\big(1 - F_i(\kappa_i)\big) &&\text{(definition of } \Delta_i) \\
&> U_{i^{opt}} + \epsilon &&\text{(choice of } A) \\
&\ge U_{i^*} + \epsilon &&(i^{opt} \text{ is optimal}) \\
&\ge LB_{i^*} + \epsilon &&\text{(Lemma 1)}
\end{aligned}
$$

so the skeptic is not convinced that $i^*$ is $\epsilon$-optimal. $\qquad\square$

**Theorem 3** With probability at least $1 - \delta$ the Naive procedure returns an $\epsilon$-optimal algorithm. It takes enough $\kappa$-capped runtime samples of each algorithm to ensure that $2\sqrt{\frac{\ln(2n/\delta)}{2m}} + u(\kappa) \le \epsilon$.

*Proof.* The number of samples is immediate from Naive's definition of $m$. Lemma 4 says that the good events $\big\{\widehat{U}_i - \frac{\epsilon + u(\kappa)}{2} \le U_i \le \widehat{U}_i + \frac{\epsilon - u(\kappa)}{2}\big\}$ hold simultaneously for all $i$ with probability at least $1 - \delta$:

$$
\begin{aligned}
\widehat{U}_i - \frac{\epsilon + u(\kappa)}{2} &\le \widehat{U}_i - \sqrt{\frac{\ln(2n/\delta)}{2m}} - u(\kappa) &&\text{(choice of } m) \\
&\le U_i &&\text{(Lemma 4)} \\
&\le \widehat{U}_i + \sqrt{\frac{\ln(2n/\delta)}{2m}} &&\text{(Lemma 4)} \\
&= \widehat{U}_i + \frac{\epsilon - u(\kappa)}{2} &&\text{(choice of } m).
\end{aligned}
$$

If the good events hold we will have

$$
\begin{aligned}
U_{i^*} &\ge \widehat{U}_{i^*} - \frac{\epsilon + u(\kappa)}{2} &&\text{(good events)} \\
&\ge \widehat{U}_{i^{opt}} - \frac{\epsilon + u(\kappa)}{2} &&\text{(choice of } i^*) \\
&\ge U_{i^{opt}} - \epsilon &&\text{(good events).}
\end{aligned}
$$

So with probability at least $1 - \delta$, the returned algorithm is $\epsilon$-optimal. $\qquad\square$

**Theorem 4:** With probability at least $1 - \delta$ UP eventually returns the optimal algorithm and it returns an $\epsilon$-optimal algorithm if it is run until $m$ is large enough that $2\sqrt{\frac{\ln(11nm^2(\log \kappa_{i^{opt}} + 1)^2/\delta)}{2m}} + u(\kappa_{i^{opt}})\big(1 - F_{i^{opt}}(\kappa_{i^{opt}})\big) \le \epsilon$. For any suboptimal $i$, if $m, \kappa_i, \kappa_{i^{opt}}$ are ever large enough that $2\sqrt{\frac{\ln(11nm^2(\log \kappa_i + 1)^2/\delta)}{2m}} + 2\sqrt{\frac{\ln(11nm^2(\log \kappa_{i^{opt}} + 1)^2/\delta)}{2m}} + u(\kappa_i)\big(1 - F_i(\kappa_i)\big) + u(\kappa_{i^{opt}})\big(1 - F_{i^{opt}}(\kappa_{i^{opt}})\big) \le \Delta_i$ then $i$ will be eliminated.

*Proof.* An execution of UP will be called *clean* if for all $i$ and $m$ we have

$$
LCB_{im} \le U_i \le UCB_{im}
$$

and

$$
UCB_{im} - LCB_{im} \le 2\alpha_{im} + u(\kappa_i)\big(1 - F_i(\kappa_i)\big).
$$

For any given $i$ and $m$, Lemma 4 says that the above four inequalities hold simultaneously with probability at least $1 - \frac{\delta}{\frac{11}{4}nm^2(\log \kappa_{i^{opt}}+1)^2}$. Noting that $\sum_{r=1}^{\infty} \frac{1}{r^2} = \frac{\pi^2}{6} < \sqrt{\frac{11}{4}}$, a union bound over all $i$, $m$ and the four inequalities above says that an execution is clean with probability at least $1 - \delta$.

At any iteration $m$ during a clean execution we will have

$$UCB_{i^{opt}m} \geq U_{i^{opt}} \qquad \text{(clean)}$$
$$\geq U_{i^*} \qquad (i^{opt} \text{ is optimal})$$
$$\geq LCB_{i^*m} \qquad \text{(clean)}$$

so $i^{opt}$ is never eliminated.

If $i$ is suboptimal, and if $m, \kappa_i, \kappa_{i^{opt}}$ are large enough, then we will have

$$UCB_{im} \leq LCB_{im} + 2\alpha_{im} + u(\kappa_i)\big(1 - F_i(\kappa_i)\big) \qquad \text{(clean)}$$
$$\leq U_i + 2\alpha_{im} + u(\kappa_i)\big(1 - F_i(\kappa_i)\big) \qquad \text{(clean)}$$
$$= U_{i^{opt}} - \Delta_i + 2\alpha_{im} + u(\kappa_i)\big(1 - F_i(\kappa_i)\big) \qquad \text{(definition of } \Delta_i)$$
$$< U_{i^{opt}} - 2\alpha_{i^{opt}m} - u(\kappa_{i^{opt}})\big(1 - F_{i^{opt}}(\kappa_{i^{opt}})\big) \qquad (m, \kappa \text{ large enough})$$
$$\leq UCB_{i^{opt}m} - 2\alpha_{i^{opt}m} - u(\kappa_{i^{opt}})\big(1 - F_{i^{opt}}(\kappa_{i^{opt}})\big) \qquad \text{(clean)}$$
$$\leq LCB_{i^{opt}m} \qquad \text{(clean)}$$
$$\leq LCB_{i^*m} \qquad \text{(choice of } i^*)$$

so $i$ will be eliminated.

If we reach a point where $2\sqrt{\frac{\ln(11nm^2(\log \kappa_{i^{opt}}+1)^2/\delta)}{2m}} + u(\kappa_{i^{opt}})\big(1 - F_{i^{opt}}(\kappa_{i^{opt}})\big) \leq \epsilon$ then we will have

$$U_{i^*} \geq LCB_{i^*m} \qquad \text{(clean)}$$
$$\geq LCB_{i^{opt}m} \qquad \text{(choice of } i^*)$$
$$\geq UCB_{i^{opt}m} - 2\alpha_{i^{opt}m} - u(\kappa_{i^{opt}})\big(1 - F_{i^{opt}}(\kappa_{i^{opt}})\big) \qquad \text{(clean)}$$

so $i^*$ is $\epsilon$-optimal. $\qquad \square$

**Theorem 5:** For any $m$, let $\epsilon = 3\sqrt{\frac{\ln(11nm^4/\delta)}{2m}}$. Then with probability at least $1 - \delta$, UP returns an $\epsilon$-optimal algorithm once it takes $m$ samples. At that point, if $\kappa_i^*$ is the largest captime algorithm $i$ has been run with then $\kappa_i^* \leq 2 \inf \left\{ \kappa \ : \ u(\kappa)\big(1 - F_i(\kappa)\big) < \frac{\epsilon}{3\sqrt{2}} \right\}$.

*Proof.* Let $\alpha_m = \sqrt{\frac{\ln(11nm^4/\delta)}{2m}}$ and note $\alpha_m \leq \sqrt{2}\alpha_{im}$ for all i. Also not that since $\kappa_i$ is doubled at most once per increment of $m$, we have that $m \geq \log(\kappa_i) + 1$ and so $\alpha_{im} \leq \alpha_m$.

The doubling condition means we have had in every iteration $m$ since $\kappa_{i^{opt}}$ was last doubled

$$2\alpha_{i^{opt}m} > u(\kappa_{i^{opt}})\big(1 - \widehat{F}_{i^{opt}m}(\kappa_{i^{opt}})\big),$$

and Lemma 2 says that

$$u(\kappa_{i^{opt}})\big(1 - \widehat{F}_{i^{opt}m}(\kappa_{i^{opt}})\big) \geq u(\kappa_{i^{opt}})\big(1 - F_{i^{opt}}(\kappa_{i^{opt}})\big) - u(k_{i^{opt}})\alpha_{i^{opt}m}.$$

Combining these we have

$$2\alpha_{i^{opt}m} + u(\kappa_{i^{opt}})\big(1 - F_{i^{opt}}(\kappa_{i^{opt}})\big) \leq \big(2 + u(\kappa_{i^{opt}})\big)\alpha_{i^{opt}m}.$$

Since $u(\kappa_{i^{opt}}) \leq 1$ and $\alpha_{i^{opt}m} \leq \alpha_m$, this means that

$$2\alpha_{i^{opt}m} + u(\kappa_{i^{opt}})\big(1 - F_{i^{opt}}(\kappa_{i^{opt}})\big) \leq 3\alpha_m$$
$$= \epsilon$$

and so by Theorem 4, UP returns an $\epsilon$-optimal algorithm with probability at least $1 - \delta$.

For any $i$, let $m^*$ be the last round in which we doubled $\kappa_i$. This means that in round $m^*$ we had done $m^*$ runs of $i$ at captime $\kappa_i^*/2$, observed that $2\alpha_{im^*} \leq u(\kappa_i^*/2)\big(1 - \widehat{F}_i(\kappa_i^*/2)\big)$ and then doubled $\kappa_i$ for the last time to $\kappa_i^*$.

Since we observed that $2\alpha_{im^*} \leq u(\kappa_i^*/2)\big(1 - \widehat{F}_i(\kappa_i^*/2)\big)$, we will have had

$$
\begin{aligned}
u(\kappa_i^*/2)\big(1 - F_i(\kappa_i^*/2)\big) &\geq u(\kappa_i^*/2)\big(1 - \widehat{F}_i(\kappa_i^*/2)\big) - u(\kappa_i^*/2))\alpha_{im^*} && \text{(Lemma 2)} \\
&\geq \big(2 - u(\kappa_i^*/2)\big)\alpha_{im^*} && \text{(observed)} \\
&\geq \alpha_{im^*} && (u \text{ bounded}) \\
&\geq \frac{\alpha_{m^*}}{\sqrt{2}} && \text{(fact above)} \\
&= \frac{\epsilon}{3\sqrt{2}} && \text{(definition).}
\end{aligned}
$$

Thus, for every $m > m^*$ we will have $\frac{\epsilon}{3\sqrt{2}} \leq u(\kappa_i^*/2)\big(1 - F_i(\kappa_i^*/2)\big)$. Since $u(\kappa)\big(1 - F_i(\kappa)\big)$ is monotonically decreasing in $\kappa$, this means that $\kappa_i^*/2 \leq \inf\big\{\kappa \ : \ u(\kappa)\big(1 - F_i(\kappa)\big) < \frac{\epsilon}{3\sqrt{2}}\big\}$. $\qquad\square$