# OpenReview forum: "Utilitarian Algorithm Configuration"
_NeurIPS.cc/2023/Conference — NeurIPS 2023 poster_

### Official Review · Reviewer_Azei · 2023-06-21

**Soundness:** 3 good
**Presentation:** 1 poor
**Contribution:** 2 fair
**Rating:** 5
**Confidence:** 2

**Summary:**

This paper studies the algorithm configuration problem with a monotone (invertible) utility function. We are given a set of $n$ algorithms, and instances are drawn from a distribution, $\mathcal{D}$. We can observe samples of running times of $i\in [n]$ on $j\sim\mathcal{D}$ capped by $\kappa>0$, and we aim to find an $\epsilon$-optimal algorithm from $[n]$. The characteristic of the setting studied in this paper is that the cost paid for each sample depends on the time we spent collecting it and that only capped runtime can be observed.

The authors first discuss lower bounds on the number of samples and captime required for identifying an $\epsilon$-optimal algorithm (Sections 3.1 and 3.2). The result on captime was constructively proved via a prover--skeptic procedure. The authors then give a naive configuration method (Naive), which requires specifying $\epsilon$. This requirement is impractical since a slight change in $\epsilon$ may largely affect appropriate $m$ and $\kappa$ values (this is also confirmed empirically in Section 4). Section 3.4 addresses this problem and presents the Utilitarian Procrastination method (UP), an anytime algorithm that starts from the smallest captime and increases it if necessary. Experiments in Section 4 demonstrate the advantage of UP over Naive.

**Strengths:**

1. The paper studies an interesting direction of algorithm configuration based on the utility-theoretic approach of (Graham et al. 2022).
2. The setup and algorithms are well described.
3. The technical motivation of UP is described clearly. The results are fairly strong, considering the lower bounds.

**Weaknesses:**

1. The technical novelty appears to be somewhat marginal, given that the idea of procrastination is already developed.
2. The utilitarian setting, albeit interesting, looks somewhat artificial. It is not easy to imagine realistic utility functions that naturally appear in practice.
3. The lower bound in Section 3.2 seems to be considering too pessimistic inputs, which in turn makes it easier to obtain nearly tight upper bounds in the subsequent sections (although similar tightness results are often appreciated).

**Questions:**

1. Although algorithm configuration with bounded utility functions is new, I have not felt it is technically very different from the previous setting (Kleinberg et al. 2017, 2019), particularly because monotone and invertible utility functions could simplify technical treatment. I would appreciate it if the authors could describe the main technical novelty in the utilitarian setting.
2. Are there natural examples of utility functions used in practice?
3. While $u$ is assumed to be invertible in Section 2, the function used in Section 4 (i.e., $u(t) = 10/t$ for $t>10$ and $u(t) = 0$ o.w.) is not on $\\{t: u(t)=0 \\}$. Is it possible to relax the assumption to deal with such non-invertible functions? (As far as I see, the invertibility on $\\{t: u(t)>0\\}$ seems sufficient).

**Limitations:**

While not explicitly discussed, the range to which the results apply appears to be clear.

---

> ### Author Rebuttal · Authors · 2023-08-06
>
> We thank the reviewer for their time and address each of their concerns below.
>
> Weaknesses:
>
> 1. We see the strength of our procedure (UP) as being in its anytime-ness rather than in its use of procrastination. In our context, "procrastination" really just means capping runs at small captimes before trying again later with larger ones. This is essentially necessary to make any theoretical guarantees, since the alternative would be to let some runs run for an unbounded amount of time. See Question 1 below for more discussion on technical novelty.
>
> 2. See our response to Question 2 below.
>
> 3. We wanted to state a result that parallels the classic sample complexity result described in Section 3.1. Considering only sample complexity, the worst-case number of samples required grows with $1/\epsilon^2$, but the tighter "input-dependent" bound allows a number of samples that grows with $1/\Delta^2$ for suboptimal algorithms, where $\Delta$ is the sub-optimality gap. When algorithms are very suboptimal, and $\Delta \gg \epsilon$, these bounds will be very different. Our result in Section 3.2 is also "input-dependent" in that it scales with $\Delta$ as well, but also with $F(t)$, the algorithm's runtime CDF. If $\Delta$ is very large, the captime $\kappa$ required to satisfy Lemma 7 will be smaller, since the inequality will be easier to satisfy. Similarly, for runtime distributions that are likely to complete before the captime $\kappa$, the term $1-F(\kappa)$ will be very small and will tend to be less than $\Delta + \epsilon$ no matter how small $\Delta$ is. In this way, the captime bound described in Section 3.2 is analogous to the well-established sample-complexity bound described in Section 3.1.
>
>
>
> Questions:
>
> 1. We see our novel technical contributions as including the following:
>
>     (1). We present a configuration procedure (UP) with performance guarantees that is anytime in the optimality guarantee it makes with respect to both the number of samples $m$ and the captime $\kappa$. Growing $m$ and $\kappa$ simultaneously in a way that does not cause the configuration procedure to incur a large runtime cost is not straightforward. UP balances the growth of these in a way that ensures it will eventually find the optimal algorithm by gradually ruling out all suboptimal ones, a guarantee that existing procedures like those in Kleinberg et al. (2017, 2019) do not (can not) make.
>
>     (2). We show an input-dependent lower bound on the captime required by any configuration procedure (Lemma 7, Section 3.2). To our knowledge this has not been shown elsewhere and is a useful contribution as it provides a non-trivial performance target for actual configuration procedures and a benchmark against which to compare them. While the basic idea behind the proof of Lemma 7 is simple (we do not know what things will look like beyond the captime $\kappa_i$; the algorithm could finish immediately after, or it could run forever), the proof relies on the construction of a counterexample that is non-obvious in a subtle way: it holds for any value of $F_i(\kappa_i)$ and so it does not rely on the skeptic being fooled only after having seen a particular value of $F_i(\kappa_i)$ (i.e., it holds no matter how many uncapped runs the skeptic sees).
>
>     (3). We use a cleaner and more intuitive notion of optimality than runtime-based works such as Kleinberg et al. (2017, 2019). The theoretical guarantees of Kleinberg et al. (2017, 2019) and other similar procedures need to be defined with respect to an additional parameter, which our work is able to eliminate. The guarantee we make is of the form "here is an algorithm whose performance is within $\epsilon$ of the optimal algorithm", while the guarantee made in previous works is of the form "here is an algorithm whose performance is within $\epsilon$ of the optimal algorithm *if we ignore the $\delta$-fraction of each algorithm's longest runs*".
>
>
> 2. Graham et al. (2022) give some concrete examples of utility functions that could be used in practice and give arguments about how practitioners might go about identifying utility functions that describe their own preferences. If we have clear costs of compute, we know something about how much time we will have to run the algorithm, and we can specify a value we obtain from solutions that the algorithm provides, then we can specify a definite utility function. This will essentially be a linear transformation of the expected profit that we earn. Another natural utility function used in practice is the fraction of runs completed before a certain captime. Beyond this, while utility functions are not yet commonly seen in the AC/algorithmics community, they are a ubiquitous feature in economics and decision theory. It is well accepted that individuals prefer financial outcomes according to their utility, not their raw monetary value, and Graham et al. (2022) have argued that the same should be so of algorithms and their runtimes. Though specifying them in practice may sometimes be challenging, we believe it is clear that optimizing utility functions rather than raw runtimes is the right thing to do in a fundamental way.
>
>
> 3. The function $u(t)$ does not really need to be invertible anywhere. Defining $u^{-1}(x) = \inf ( t : u(t) \le x ) $ is really what is needed. Since $u(t)$ must be monotonically decreasing this intuitively just means that $u^{-1}(x)$ is the first (i.e., smallest) runtime where our utility falls below a given level $x$. We will change this in the paper.

---

> > ### Comment · Reviewer_Azei · 2023-08-14
> >
> > I appreciate the authors' response, particularly about the difference from (Kleinberg et al. 2017, 2019). Given the above clarification (as well as the comments by Reviewer Sepd, who seems to be more familiar with algorithm configuration), I increase my score from 4 to 5.

---

### Official Review · Reviewer_3kwZ · 2023-07-03

**Soundness:** 4 excellent
**Presentation:** 4 excellent
**Contribution:** 3 good
**Rating:** 7
**Confidence:** 4

**Summary:**

The paper presents the first nontrivial procedure for configuring heuristic algorithms to  maximize the utility provided to their end users while also offering theoretical guarantees about performance. The paper shows how to avoid extremely long runs with low probability by putting limits on the runtime, assuming a bounded utility that is monotonically decreasing in runtime. The paper compares several ideas
(1) the idea behind the runtime Oracle Procedure that eliminates algorithms progressively;
(2) the idea of using a time limit whose value doubles until the optimal algorithm is identified (this idea is also used in restart procedures)
(3) estimations of the mean time and the fraction of the distribution that has been explored

The results shows that the proposed procedure returns an optimal algorithm with a certain probability with a number of samples. It also characterizes the upper bound on the threshold. The results are with a logarithmic factor of optimality for the samples and provide a constant bound for the time limit.

**Strengths:**

- novel algorithms with performance guarantees
- well-written and intuitive paper
- automatic tuning of the time limit


**Weaknesses:**

- experimental results could be stronger and use a lot more problems

**Questions:**

no question

**Limitations:**

yes, the authors mention that the bounds are not tight yet.

---

> ### Author Rebuttal · Authors · 2023-08-06
>
> We thank the reviewer for their time. We note only that we are looking at much more extensive experiments in future work.

---

> > ### Comment · Reviewer_3kwZ · 2023-08-11
> > **Rebuttal**
> >
> > I read the rebuttal and I am fine with this paper.

---

### Official Review · Reviewer_Sepd · 2023-07-05

**Soundness:** 3 good
**Presentation:** 3 good
**Contribution:** 4 excellent
**Rating:** 7
**Confidence:** 3

**Summary:**

Theoretical guarantees for algorithm configuration (AC) when using a utilitarian objective function are proposed. This work differs from previous work in several ways, the first being that it focuses on the "utilitarian" objective function, which differs from the objective functions used in essentially all previous work. Second, the algorithm proposed provides anytime theoretical guarantees, meaning key parameters from previous approaches need not be input. A short experimental evaluation is performed on some well-known datasets for examining theoretical AC approaches to further drive home the results from the theorems.

**Strengths:**

1. The paper is well-organized and clearly written.

2. The introduction of an anytime approach for AC with theoretical guarantees is a rather significant result. One of the key drawbacks of previous work is that the decision maker must choose values like epsilon and kappa that have little to no meaning to a non-expert. Integrating these values into the approach itself is a very welcome advancement that will have an impact on future work in this area.

3. I am not sure whether the use of the utilitarian objective function is a strength or weakness, it has both aspects to it, so I include it here. On the one hand, the paper bases all of its work on the utilitarian objective function, which has not caught on with the empirical side of AC yet. Perhaps it will. At any rate, while it offers a way forward in terms of the proofs that was potentially unavailable before, it also somewhat strands this work away from the rest of the AC literature, making comparisons difficult and it somewhat hard to assess exactly where this work stands. I think overall the utilitarian objective function is a net positive, hence I place this comment under strengths.

**Weaknesses:**

1. I found the proofs both in the paper and in the supplementary material (although I admit I have not fully checked the supplementary material) somewhat... let's call it almost arrogant? I am sorry to use that word, but for less mathematically inclined readers, I think they will come away from this paper feeling rather stupid. I know I did.

2. Regarding lemma 1, and potentially putting myself in the category of stupid reviewers, I do not understand how U_i <= U_i(kappa). I just do not understand how the uncapped expected utility could possibly be less than the capped utility. Clearly, they can in some cases be equal. But is this a typo? This is an example where an actual proof would lead to better understanding.

3. I am including the dataset used as a weakness, because a better dataset is sorely needed for comparing theoretical approaches. I do note that the dataset used is better than nothing, and this point does not influence my score, as the authors practically have no choice other than to expend significant computational resources.

4. I would have appreciated a more detailed explanation of utilitarian AC. The reader really cannot be expected to know about or have read the previous paper in detail to evaluate this one. In the future, this would serve as a refresher for those who have read the utilitarian paper and need to be reminded.

5. The authors provide overall a good justification for not comparing to previous techniques, I do wonder, though, what the performance would be like if one modeled a utility function after the functions usually used -- would this not be possible?

6. The authors left out the most recent theoretical approach, AC-Band, in the related work, but this is not important for my evaluation.

**Questions:**

1. Relating to weakness #5, would it not be possible to craft a utility function close to, say, PAR10 or mean runtime for comparison purposes?

2. In AC, mean runtime and PAR10 have become standard functions because (1) they are easy to compute and (2) they relatively closely reflect what algorithm users actually want. Long runtimes are indeed reduced by PAR10, even if the 10 is very arbitrary. If I do not have many timeouts, is the utility function still "better"? Are there some clear use cases that can be described?

**Limitations:**

The limitations are adequately discussed within the paper.

---

> ### Author Rebuttal · Authors · 2023-08-06
>
> We thank the reviewer for their time and address each of their concerns below.
>
> Weaknesses:
>
> 1. We plead ignorance not arrogance. Being so familiar with the work it becomes hard to see it from the perspective of someone who is not, and easy to forget that what seems obvious now was not always so. We will endeavor to make explanations of the proofs more straightforward and clear.
>
> 2. See 1. above. The capped expected utility $U_i(\kappa)$ could be greater than the uncapped expected utility if there is any chance of a run exceeding the captime $\kappa$. This is because the utility function $u$ is monotonically decreasing, and so if runtime exceeds $\kappa$ we will have $u(\kappa) \ge u(runtime)$. Intuitively, the capped expected utility counts capped runs as having just completed at the captime, when really they would have taken longer if given the chance. Since we prefer short runs, this means that the capped expected utility looks better than the uncapped. Again, we will make this explanation more clear in the paper.
>
> 3. We note only that there are actually three datasets (albeit from a single source) and that we are working on much more extensive experiments.
>
> 4. We will include more explanation of utilitarian AC. Specifically, we can better explain what motivates the use of utility functions, what assumptions are needed to ensure the existence of utility functions, and what is implied about the form of those functions.
>
> 5. Graham et al. (2022) have argued that many scoring functions used in practice are not consistent with a certain form of rational decision making. Some functions (such as PAR10) could be expressed as functions of this form if a consistent capping strategy is included. For example, if we cap all runs at 100 seconds and use a PAR10 scoring function then this is equivalent to what Graham et al. (2022) have called a linear utility function. It is linear from 0 to 100 seconds, with $u(0) = 1$ and  $u(100) = 1 - 100/1000$, and $u(t) = 0$ for $t$ greater than 100 seconds. (To derive this we apply a linear transformation to the PAR10 score: multiply the PAR10 score by -1 and add 1000, then divide this by 1000.) Certainly utility functions of this sort could be used.
>
> 6. We will include an explanation of AC-Band.
>
> Questions:
>
> 1. See response to Weakness 5 above. Graham et al (2002) include discussion of the mean runtime objective, arguing that it is a poor formalization of algorithm users' preferences for various reasons. We do not repeat that full argument here, but some of the reasons include that it essentially assumes that we obtain no increase in happiness for solving our problem and that it doesn't coincide with actual behavior. If practitioners did care about average runtime, they would be unable to compare algorithms that occasionally fail to complete, since both averages would be infinite.
>
> 2. We think of the utility function as something that exists *a priori* and reflects the user's preferences, rather than something that can be used to improve optimization performance (although it is nice that this can also be the case). As discussed in the response to Weakness 5 above, there are ways in which PAR10 (or more generally PARk) scoring functions are directly equivalent to some utility function $u(t)$ of the form we have assumed. But the converse is not always true. The idea behind using more general utility functions is that they can more accurately express the true preferences of users. As pointed out, the 10 is arbitrary. What's more, the point at which we choose to apply the x10 penalty (i.e., the captime) is also somewhat arbitrary. Referring back to the response to Weakness 5 above, maybe we want a x10 penalty at captime 100 seconds, but also a x2 penalty at captime 50 seconds. There is a $u(t)$ that can reflect this preference. With this interpretation, there is no sense in which one utility function is "better" than another, they simply reflect preferences.

---

> > ### Comment · Reviewer_Sepd · 2023-08-11
> > **Rebuttal**
> >
> > I read the rebuttal and thank the authors for clearing things up for me. I stand by my score that this paper should be accepted.

---

### Official Review · Reviewer_Qzq9 · 2023-07-06

**Soundness:** 1 poor
**Presentation:** 4 excellent
**Contribution:** 1 poor
**Rating:** 2
**Confidence:** 5

**Summary:**

The paper introduces a setting and an algorithm for algorithm configuration based on utility of the algorithm's running time. Theoretical guarantees are provided and upper and lower bounds are compared against. The algorithm is evaluated on case studies.

**Strengths:**

Utility-based decision making approach is applied to algorithm configuration. The proposed approach is theoretically analyzed and empirically evaluated.

**Weaknesses:**

## Literature

On time (or resource in general) constrained utility based decision making, start with this seminal work by Stuart Russell and his PhD student Eric Wefald: https://mitpress.mit.edu/9780262513821/do-the-right-thing/

For a   more updated recent account on bounded rationality (that is, on making utility based decisions under computational resource constraints) look at this work (UAI 2012): https://arxiv.org/abs/1207.5879

Eric Horvitz is the 'father' of resource-bounded decision making.  Citations on bounded rationality are here: http://erichorvitz.com/computational_rationality_readings.htm. A good collection of articles coedited by Eric Horvitz and Shlomo Zilberstein is in this special issue of AIJ: https://www.sciencedirect.com/journal/artificial-intelligence/vol/126/issue/1

## Problem with the submission under discussion

An algorithm is selected based on its expected net utility. The net utility consists of two terms, the intrinsic utility and the deliberation cost. Relevant work on algorithm configuration assumes that all algorithms have the same intrinsic utility (solve the problem 'exactly' or with solutions of indistinguishable quality), and their total utility depends on deliberation cost --- which is in many settings just the running time.

Intuitively, the intrinsic utility is what you get and the deliberation cost is what you pay. The theory of utility to which the authors refer applies to the intrinsic utility.  The computation/time cost is an additive negative term in the total utility, which conforms to different laws, see the citations above for detailed discussion.

The authors confuse the intrinsic utility and the time cost, and apply non-linear bounding transformation to the running time instead of to a parameter of the solution quality. They then come up with an algorithm that allows to select algorithms more efficiently than the baseline based on the time cost. However, due to inherent inconsistency in applying bounding transformation to the time cost, the authors' approach results in a selection/configuration scheme that may prefer an algorithm that does not always terminate to an algorithm that terminates, even within reasonable time limits. This is just an extreme example.

As another intuition, assume there are two luxury car models, one costs USD1 million, the other USD2 million. The authors argue that USD1 million is such a huge sum that USD1 million or USD2 million do not differ much, and if you have USD1000000 but want the USD2000000 car, you **can just add some pocket change** to buy it. However, this will most probably not work. To get a USD2000000 car you still need twice amount of money you would pay for USD1000000 car, the amount of money you can buy **2 USD1000000 cars**.  However, the value you get from a USD2000000 car, whether for vanity or practicality, is not twice the value of USD1000000 car, it is likely to be the same or just a little bit higher (unless there is a club of USD2000000 car owners you want badly to be accepted to).
Here, the car price is the cost (deliberation cost, time cost in algorithm selection) and the value is the intrinsic utility (how good the algorithm solution is).

The problem in the submission is in improper use of utility-based decision making to select computations. The theory is detailed in the works cited above.


**Questions:**

Is this the net or the intrinsic utility?

**Limitations:**

The limitations are addressed adequately.

---

> ### Author Rebuttal · Authors · 2023-08-06
>
> We thank the reviewer for their time, unfortunately we have had a hard time understanding their objections. We have tried our best to address each of their concerns below.
>
> "The approach is based on a notion of utility that is not consistent neither with the common practice of utility functions, nor with utility based decision making theory, nor with the notion of rationality." - The existence of the utility function we use was proved in Graham, et al. (2022), peer reviewed and published work. This utility function is based on the Von Neumann-Morgenstern (1947) axiomatization, which is an extremely well-established notion of "rationality" and has been a cornerstone of expected utility-based decision theory for three quarters of a century. We accept that there are other functions called "utility functions" (e.g., quasilinear) but these are not the functions we are considering. The axioms of Graham, et al. (2022) imply a certain class of utility function, and we assume a function of this form. Whether one finds those axioms to be consistent with their idea of "rationality" is a personal choice, but it is one that is very commonly made in other works and certainly leads to a sensible and consistent notion of utility.
>
> "For decision making the utility must be bounded above, but it does not have to be bounded from below..." - VNM utility functions are bounded from above and from below.
>
> "The proposed notion of probability bounded between 0 and 1 is not just 'impractical', it leads to contradictions in rational decision making..." - Probabilities, of course, are bounded between 0 and 1 by definition, so we are not sure what is meant by this comment.
>
> "If, however, the notion of utility is made consistent with the decision theory, the proposed approach cannot be constructed, as far as I understand." - Given that our notion of utility *is* consistent with VNM-based decision theory, as shown in Graham et al. (2022), and that we *have* constructed the proposed approach, UP, we would suggest that the reviewer has indeed misunderstood something. We hope that the above explanations have helped to clarify this.
>
> "Is this the net or the total utility?" - The function $u(t)$ represents the utility gained from a single run that takes $t$ seconds to complete.

---

> > ### Comment · Reviewer_Qzq9 · 2023-08-12
> > **Clarifications of objections.**
> >
> > 1. The utility can be bounded if defined on bounded (discrete or continuous) domain, but not the way you introduce it.
> >
> > 2. Consider two algorithms. Algorithm A always terminates after 5 seconds. Algorithm B terminates after 0 seconds (successfully) with probability 0.5, otherwise does not terminate at all. Let us assume that your utility function assings the utility of 1 to 0, 0.1 to 5, and zero to infinity. According to your approach, algorithm B has higher expected utility (0.5) than algorithm A (0.1), which I find to be contradictory.  An algorithm that does not terminate with non-zero probability should have a lower uitlity than an algorithm that always terminates.
> >
> > 3. Rationalilty is not a matter of personal feelings. There is an extensive body of work on rationality and utility. Look at Eric Horvitz's and Stuart Russell's work, for example.
> >
> > 4. Net utility and total utility are notions of decision making under uncertainty, sorry. I am asking whether u(t) is net or total. University textbooks on AI, such as by Stuart Russell, Dave Poole etc. explain this difference. Roughly, net utility is the benefit you get from the result. Total is the benefit you get from the result less the cost to obtain the result. You didn't answer my question.
> >
> > Based on the authors' feedback, I have updated my score. It is my impression that the authors are not sufficiently familiar with the literature and with the utility theory , and this causes critical flaws in the paper.

---

> > > ### Author Response · Authors · 2023-08-13
> > > **Further clarification**
> > >
> > > We believe that the reviewer has certain misunderstandings about the notions of utility and rationality. Nevertheless, we respond to their comments below.
> > >
> > > 1. "The utility can be bounded if defined on bounded (discrete or continuous) domain, but not the way you introduce it." - The reviewer seems to be claiming that only functions on bounded intervals can be bounded, which is certainly not the case. Regardless, we reiterate that the purpose of our work is not to argue the existence of the utility function. That has been proved in previously published work [Graham et al. 2022]. The purpose of the present work is only to use the utility function, which has been guaranteed to exist.
> > >
> > > 2. When the reviewer writes statements like "... which I find to be contradictory" or "an algorithm that does not terminate with non-zero probability should have a lower utility than an algorithm that always terminates" they are making normative statements. The whole point of this work is to avoid such statements, accept a user's preferences as given (whatever they may be --- who are we to say?), and provide them with the best-possible guidance given their preferences. In reference to the given example, a user who needs solutions to their problem in exactly 4 seconds gains nothing from Algorithm A and so should clearly prefer Algorithm B. There is nothing contradictory about this.
> > >
> > > 3. There are many widely-used but distinct definitions of "rationality," depending on the properties desired of the definition (e.g., reason-responsiveness vs. internal coherence). See, e.g., the Wikipedia page on rationality to get a quick sense of this. We use the (extended) VNM axioms of Graham et al. (2022) as our definition which, again, is a very well-established notion. Indeed, we note that both Russell and Poole rely on the VNM framework as well.
> > >
> > > 4. The function $u(t)$ is well-defined as is, and represents whatever it is the decision-maker might care about (e.g., including costs in decision-making or not). One might speculate that many decision-makers' utility functions would incorporate costs incurred ("net" in the usual sense), but the framework of the paper does not require this.

---

> > > > ### Comment · Reviewer_Qzq9 · 2023-08-13
> > > >
> > > > Can you provide a practical example of a problem in which a non-terminating algorithm has a higher utility than a terminating one?

---

> > > > > ### Author Response · Authors · 2023-08-13
> > > > >
> > > > > Suppose we make deliveries each day as part of our business. Each morning we get a list of locations that we need to visit for that day. We want to try and route our vehicles efficiently and we have two possible algorithms. Algorithm A always solves every routing problem optimally in exactly 25 hours. Algorithm B solves half of the routing problems optimally in 1 hour and fails to terminate on the other half. Although Algorithm A always terminates, it is of no use to us since it never terminates in time (i.e., we need to make our deliveries today). On the other hand, Algorithm B can at least solve half of the problem instances quickly. For the other half we can follow some default behaviour (e.g., visit the stops in alphabetical order) and still be better off on average than if we had used Algorithm A.

---

> > > > > > ### Comment · Reviewer_Qzq9 · 2023-08-14
> > > > > >
> > > > > > 1. You describe here two algorithms that terminate.
> > > > > >
> > > > > >   * Algorithm A runs for 25 hours and terminates, returning the optimal solution.
> > > > > >   * Algorithm B either runs for 1 hour and terminates returning the optimal solution, or is stopped after some time, and then returns a suboptimal solution.
> > > > > >
> > > > > > 2. This setting has a property not addressed in your paper: the utility of algorithm A at t=25 is higher than the utility of algorithm B at t=25 (because the former returns the optimal solution and the latter returns a suboptimal solution). In your work, all algorithms share the same utility function. One way to take care of the latter is to discern between intrinsic and net utilities, which your submission does not do.

---

> > > > > > > ### Author Response · Authors · 2023-08-15
> > > > > > >
> > > > > > > Note that all algorithms *must* share the same utility function, or we would not be able to compare them according to their expected utility (just as we cannot compare bundles of economic goods in a meaningful way by applying a different utility function to each bundle).
> > > > > > >
> > > > > > > As stated previously, the whole point of this work is to simply accept a user's utility function as given. In the example, we would imagine the value of the user's utility function to be 0 at t=25 because learning a solution at that time (optimal or otherwise) is of absolutely no use, being past the deadline. More generally, a user's utility function can incorporate a notion of solution quality (such as the suboptimality gap, this is discussed in Graham et al. (2022)), though it certainly does not need to, and there is a large body of work that considers only the pure runtime case. Either way, it was not our intention to focus on this in the given example, since the way a user feels about trading longer runs for better-quality solutions is simply another personal decision.

---

> > > > > > > > ### Comment · Reviewer_Qzq9 · 2023-08-15
> > > > > > > >
> > > > > > > > I fail to understand how your comment makes your example valid. It still describes two algorithms that terminate. In addition you seem to be limiting now the family of utility functions to those which are zero at the maximum running time of an algorithm, while in your paper you only require that it approaches zero at infinity.

---

> > > > > > > > > ### Comment · Reviewer_Qzq9 · 2023-08-15
> > > > > > > > >
> > > > > > > > > Taking it in a slightly different direction, the negated running time of an algorithm is a very reasonable utility function, and this is the reason why it is used in many algorithm configurations. However, your proposed approach explicitly excludes this very reasonable utility broadly used in algorithm configuration from the range of allowed utilities by requiring that u(t) be lower-bounded.

---

> > > > > > > > > > ### Author Response · Authors · 2023-08-15
> > > > > > > > > >
> > > > > > > > > > Graham et al. (2022) argue that negative runtime, which is ruled out by the VNM axiomatization, is actually not a reasonable utility function because it implies that we care linearly about runtime, even for very long runs. Is a run that takes 100 years ten times worse than a run that takes 10 years? Is a run that takes 1000 years ten times worse again? Maybe. But maybe they are all just so long that they all give us essentially 0 utility. Either way, the axiomatization rules out utility functions that are linear over the whole positive real line and we believe, based on arguments like the above, that this is very reasonable.

---

> > > > > > > > > > > ### Comment · Reviewer_Qzq9 · 2023-08-15
> > > > > > > > > > >
> > > > > > > > > > > Intrinsic utility and net utility are not the same. The total utility is the intrinsic utility less the cost. The cost is the time. Intrinsic utility which is linear on a parameter is problematic, but the total one is a standard setting. Time cost is fine for subtracting from the negative utility. Negative time cost is a fine utility if only optimal solution is returned.

---

> > > > > > > > > ### Author Response · Authors · 2023-08-15
> > > > > > > > >
> > > > > > > > > We take "terminate" to mean "stops computation and returns the optimal solution" not "we killed the process and queried it for its current result". If the latter is meant, we would argue that every algorithm always "terminates" on every input. If the notion of solution quality is making this hard to see, we could imagine solving SAT problems in the given example instead of routing problems. In that case, the answer we get is a binary "yes/no" telling us if the formula is satisfiable or not, and quarrying the algorithm midway through computation does not tell us anything about the final answer. To get the answer, we simply have to wait until it finishes.
> > > > > > > > >
> > > > > > > > > The utility function that is zero at t=25 was given as a single specific example of what might be a reasonable utility function in the given setting. Nowhere was it claimed to be required.

---

> > > > > > > > > > ### Comment · Reviewer_Qzq9 · 2023-08-15
> > > > > > > > > >
> > > > > > > > > > This contradicts your own example in which the algorithm is stopped and a fallback default solution is returned.

---

> > > > > > > > > > > ### Author Response · Authors · 2023-08-15
> > > > > > > > > > >
> > > > > > > > > > > An algorithm run that "terminates" stops computation and returns an answer. A run that does not terminate in this sense may be stopped at any time by the user, at which point it may or may not provide the user with some intermediate value. The "default" we referred to is the action the user decides to take when they fail to get a useable answer. Nothing in this is contradictory.

---

> > > > > > > > > > > > ### Comment · Reviewer_Qzq9 · 2023-08-15
> > > > > > > > > > > >
> > > > > > > > > > > > Taking your SAT problem solver as an example, what default action should the user provide if the algorithm was stopped by the user?

---

> > > > > > > > > > > > > ### Author Response · Authors · 2023-08-15
> > > > > > > > > > > > >
> > > > > > > > > > > > > The default action that a user chooses will depend on their particular setting. Some possible examples:
> > > > > > > > > > > > >
> > > > > > > > > > > > > If the user is a chip producer, and the SAT formula is an encoding of the correct behaviour of the chip, but the producer has been unable to verify the design after running a solver for a long time, then a default action might be to scrap the design and try another one.
> > > > > > > > > > > > >
> > > > > > > > > > > > > If the user is a large manufacturer, and the SAT formula represents the resource requirements of a new job, but after running a solver for a long time the manufacturer is unable to confirm that the job can be completed at some given factory, then the default action might be to schedule the job at some larger factory where they know it can be completed.

---

> > > > > > > > > > > > > > ### Comment · Reviewer_Qzq9 · 2023-08-15
> > > > > > > > > > > > > >
> > > > > > > > > > > > > > this is entirely unrelated to the paper and the example, sorry. we are talking about a SAT solver, not about applications. There are, basically, two kinds of problems solved by algorithms: decision and optimization. SAT is  a decision problem, it does not have a satisfying candidate solution. There are optimization problems,such as MAXSAT, but this is not the example you gave.
> > > > > > > > > > > > > >
> > > > > > > > > > > > > > In general, I am afraid this discussion stopped to be productive, sorry.
> > > > > > > > > > > > > >
> > > > > > > > > > > > > > You confused intrinsic utility and time cost and applied the theory developed for intrinsic utility to time cost. I pointed at this as a major problem in your submission. This cannot be fixed through either discussion here or a minor revision, sorry.
> > > > > > > > > > > > > >
> > > > > > > > > > > > > > You cannot just pass the time cost through a sigmoid and pretend that gives a better decision making rule, it does not work this way

---

> > > > > > > > > > > > > > > ### Comment · Area_Chair_2iVp · 2023-08-15
> > > > > > > > > > > > > > > **Unproductive discussion**
> > > > > > > > > > > > > > >
> > > > > > > > > > > > > > > I agree the discussion has become unproductive. I also have to agree with the authors that I find the review itself and the reviewers’ comments after it to be hard to follow. If the reviewer would like to rewrite the review in a clear way, that would be appreciated. Also, if you want to refer to prior work, please include a citation so we can look into the work and not just say things like “Look at Eric Horvitz's and Stuart Russell's work, for example.”

---

> > > > > > > > > > > > > > > > ### Comment · Reviewer_Qzq9 · 2023-08-15
> > > > > > > > > > > > > > > > **Utility, rationality, and utility based decision making**
> > > > > > > > > > > > > > > >
> > > > > > > > > > > > > > > > ## Literature
> > > > > > > > > > > > > > > >
> > > > > > > > > > > > > > > > On time (or resource in general) constrained utility based decision making, start with this seminal work by Stuart Russell and his PhD student Eric Wefald: https://mitpress.mit.edu/9780262513821/do-the-right-thing/
> > > > > > > > > > > > > > > >
> > > > > > > > > > > > > > > > For a   more updated recent account on bounded rationality (that is, on making utility based decisions under computational resource constraints) look at this work (UAI 2012): https://arxiv.org/abs/1207.5879
> > > > > > > > > > > > > > > >
> > > > > > > > > > > > > > > > Eric Horvitz is the 'father' of resource-bounded decision making.  Citations on bounded rationality are here: http://erichorvitz.com/computational_rationality_readings.htm. A good collection of articles coedited by Eric Horvitz and Shlomo Zilberstein is in this special issue of AIJ: https://www.sciencedirect.com/journal/artificial-intelligence/vol/126/issue/1
> > > > > > > > > > > > > > > >
> > > > > > > > > > > > > > > > ## Problem with the submission under discussion
> > > > > > > > > > > > > > > >
> > > > > > > > > > > > > > > > An algorithm is selected based on its expected net utility. The net utility consists of two terms, the intrinsic utility and the deliberation cost. Relevant work on algorithm configuration assumes that all algorithms have the same intrinsic utility (solve the problem 'exactly' or with solutions of indistinguishable quality), and their total utility depends on deliberation cost --- which is in many settings just the running time.
> > > > > > > > > > > > > > > >
> > > > > > > > > > > > > > > > Intuitively, the intrinsic utility is what you get and the deliberation cost is what you pay. The theory of utility to which the authors refer applies to the intrinsic utility.  The computation/time cost is an additive negative term in the total utility, which conforms to different laws, see the citations above for detailed discussion.
> > > > > > > > > > > > > > > >
> > > > > > > > > > > > > > > > The authors confuse the intrinsic utility and the time cost, and apply non-linear bounding transformation to the running time instead of to a parameter of the solution quality. They then come up with an algorithm that allows to select algorithms more efficiently than the baseline based on the time cost. However, due to inherent inconsistency in applying bounding transformation to the time cost, the authors' approach results in a selection/configuration scheme that may prefer an algorithm that does not always terminate to an algorithm that terminates, even within reasonable time limits. This is just an extreme example.
> > > > > > > > > > > > > > > >
> > > > > > > > > > > > > > > > The problem in the submission is in improper use of utility-based decision making to select computations. The theory is detailed in the works cited above.

---

> > > > > > > > > > > > > > > > > ### Comment · Reviewer_Qzq9 · 2023-08-16
> > > > > > > > > > > > > > > > >
> > > > > > > > > > > > > > > > > As another intuition, assume there are two luxury car models, one costs USD1 million, the other USD2 million. The authors argue that USD1 million is such a huge amount of money that USD1 million or USD2 million do not differ much, and if you have USD1000000 but want the USD2000000 car, you can just _add some pocket change_ to buy it. However, this will most probably not work. To get a USD2000000 car you still need twice amount of money you would pay for USD1000000 car, _the amount of money you can buy 2 USD1000000 cars for_. However, the value you get from a USD2000000 car, whether for vanity or practicality, is not twice the value of USD1000000 car, it is likely to be the same or just a little bit higher (unless there is a club of USD2000000 car owners you want badly to be accepted to). Here, the car price is the cost (deliberation cost, time cost in algorithm selection) and the value is the intrinsic utility (how good the algorithm solution is).

---

> > > > > > > > > > > > > > > > > > ### Comment · Reviewer_Sepd · 2023-08-16
> > > > > > > > > > > > > > > > > >
> > > > > > > > > > > > > > > > > > Another reviewer butting into the discussion here.
> > > > > > > > > > > > > > > > > >
> > > > > > > > > > > > > > > > > > I fully admit I did not understand this implication of the utility function.
> > > > > > > > > > > > > > > > > >
> > > > > > > > > > > > > > > > > > Applying this logic to the runtime setting of algorithm configuration: 20 hours is not much worse than 10 hours is total nonsense. It's twice as worse. If anything above 10 hours was not interesting to me, I would have set the timeout to 10 hours and not bothered with anything longer anyway. So I do not see the value of the utility function in this case.
> > > > > > > > > > > > > > > > > >
> > > > > > > > > > > > > > > > > > If I apply the logic to quality maximization, it does not make sense either, since many scenarios involve costs being minimized or profit being maximized. Why would I want to discount a dollar earned on one instance versus another just because of the size of the objective function? That makes no sense to me.
> > > > > > > > > > > > > > > > > >
> > > > > > > > > > > > > > > > > > If my interpretation here is correct, then I am not as positive about this paper...

---

> > > > > > > > > > > > > > > > > > > ### Author Response · Authors · 2023-08-18
> > > > > > > > > > > > > > > > > > >
> > > > > > > > > > > > > > > > > > > Thanks for devoting so much time and careful thought to considering our paper and our replies. We find it important to stress that it is both consistent with utility theory and intuitively reasonable for runtime to affect a user's utility in a nonlinear way. Of course, there are certainly settings and/or regions of a function for which utility is linear in runtime. For example, a risk-neutral user with a fixed time budget running jobs on Amazon EC2 and caring only about money spent has a linear utility function. As pointed out by one of the reviewers, so does a boundedly rational agent who experiences deliberation as a similarly separable cost. Conversely, there are other cases in which a user's utility would not change linearly with the passage of time. If an employee of a tech company runs a job on a server overnight, they may experience literally no utility difference between having the job take half the night or three quarters of the night. Similarly, in many practical use cases, an algorithm that produces an answer in 10 ms is essentially equivalent to one that produces an answer in 1 ms (rather than 10x worse); both algorithms are so fast that the time difference is irrelevant. To give a final example, if we are trying to verify a new security protocol to replace a currently deployed version of the protocol that suffers from an internally discovered vulnerability, the cost we incur will almost certainly not be linear: it will be relatively small at first, while the vulnerability is undiscovered and unexploited, but could increase quickly and drastically as the risk of exploitation grows.
> > > > > > > > > > > > > > > > > > >
> > > > > > > > > > > > > > > > > > > In reference to reviewer Sepd's example, 20 hours may indeed be twice as bad as 10 hours, but is 100 hours ten times as bad? is 1000 hours 100 times as bad? There may be regions of the utility function that are linear in runtime, but does this relationship really continue out to infinity? Certainly if anything above 10 hours is not interesting to us we should set the timeout to (at most) 10 hours, but if things above 10 hours are "sort of" interesting to us, things above 20 hours are "a little" interesting to us, and things above 30 hours "just barely" interesting to us, then the picture is less clear. A similar pattern may hold for runtimes smaller than 10 hours as well. The value of using the utility function $u$ is that it is able expresses such preferences, whatever they may be.
> > > > > > > > > > > > > > > > > > >
> > > > > > > > > > > > > > > > > > > In reference to the luxury car example, we note that, empirically, individuals do not treat monetary gains or losses as affecting utility linearly either, nor does the theory of utility require that they should. Most people do not experience a loss of 2 million as being twice as bad as a loss of 1 million, and if so, do not experience a loss of 10 million as being ten times as bad (i.e., linearity does not extend out to infinity; at some point you're carrying a debt you could never repay and so don't care deeply about the exact amount). In fact, it was this observation that led to the initial development of expected utility theory.

---

> > > > > > > > > > > > > > > > > > > > ### Author Response · Authors · 2023-08-18
> > > > > > > > > > > > > > > > > > > >
> > > > > > > > > > > > > > > > > > > > Of course, the ideas just discussed are not contributions of our paper! (Instead, our paper focuses on algorithm configuration techniques for application to such settings. We are heartened that the reviewers have reacted quite positively to these technical contributions and a bit discouraged that we find ourselves in a conversation where we're having to defend related work.) More specifically, we draw our ideas about utility functions from previously published work, Graham et al. (ICML 2023), who provide a first-principles argument that $[0,1]$ bounded utility functions are able to capture preferences in the algorithm runtime setting, using a quite standard utility theoretic framework following von Neumann and Morgenstern (1947). Since we drew directly on that work, we have perhaps underemphasized antecedent ideas from the literature. For example, we reproduce an excerpt from the widely used textbook "Stochastic Local Search" (Hoos & Stützle, 2004; Google Scholar reports 2358 citations), from Chapter 4, page 157. Note that "type 1" refers to scenarios in which there are no time limits, and problems can be solved offline, "type 2" refers to scenarios where there is a hard time limit and we only want to complete the run before then, and "type 3" refers to scenarios where there is some utility function $U(t)$ that represents the usefulness of the solution at time $t$):
> > > > > > > > > > > > > > > > > > > >
> > > > > > > > > > > > > > > > > > > > "While in the case of no time limits being given (type 1), the mean run-time of a Las Vegas algorithm might suffice to roughly characterise its run-time behaviour, in real-time situations (type 2) this measure is basically meaningless. Type 3 is not only the most general class of application scenario, but these scenarios are also the most realistic. The reason for this is the fact that real-world problem solving usually involves time-constraints that are less strict than the hard deadline given in type 2 scenarios. Instead, at least within a certain interval, the value of a solution gradually decreases over time. In particular, this situation is given when taking into account the costs (in particular, CPU time) of finding a solution.
> > > > > > > > > > > > > > > > > > > >
> > > > > > > > > > > > > > > > > > > > "As an example, consider a situation where hard combinatorial problems have to be solved on line using expensive hardware in a time-sharing mode. Even if the immediate benefit of finding a solution is invariant over time, the costs for performing the computations will diminish the final payoff. Two common ways of modelling this effect are constant or proportional discounting, which use utility functions of the form $U(t) := \max\{ u_0 - c\cdot t, 0 \}$ and $U(t) := e^{\lambda \cdot t}$, respectively (see, e.g., [Poole et al., 1998])."

---

> > > > > > > > > > > > > > > > > > > ### Comment · Area_Chair_2iVp · 2023-08-18
> > > > > > > > > > > > > > > > > > > **Discounting and utilities**
> > > > > > > > > > > > > > > > > > >
> > > > > > > > > > > > > > > > > > > Reviewers, please be aware that that the authors are correct and that many or most economic models of preferences do not consider 20 hours to be "twice as bad" as 10 hours, nor is 20 dollars necessarily twice as good as 10 dollars. Below I give some examples. I suggest you update your reviews in light of the fact that linear models are often not found to be good models of human preferences. You may have other concerns about the paper, but you should be aware of the breadth of work on utility functions to model human preferences. I give these examples to illustrate that various utility functions are used to model human preferences in the field of Decision Theory and Economics and are backed up by extensive experimental research. And authors, in your next revision you could perhaps a little more background for ML reviewers who are not familiar with Economic models of utility, or other readers who are unaware of the whole field are likely to have similar reactions.
> > > > > > > > > > > > > > > > > > >
> > > > > > > > > > > > > > > > > > > # Utility for money
> > > > > > > > > > > > > > > > > > >
> > > > > > > > > > > > > > > > > > > Common utility functions for money include:
> > > > > > > > > > > > > > > > > > > 1. **Linear utility** $u(x) = x$. This models someone who is "risk neutral"
> > > > > > > > > > > > > > > > > > > 2. **Logarithmic utility** $u(x) = \log x$. This models someone who is risk averse. Someone with this utility would rather have 1,000 dollars for sure than a 50/50 gamble between 0 or 2,000 dollars.
> > > > > > > > > > > > > > > > > > > 3. **Other models** such as $u(x) = \frac{x}{x+1}$. This is just one example. Dozens of functional forms have been considered.
> > > > > > > > > > > > > > > > > > >
> > > > > > > > > > > > > > > > > > > # Discounting rewards achieved at a later time
> > > > > > > > > > > > > > > > > > >
> > > > > > > > > > > > > > > > > > > In Economics, there are a variety of models of the value of a delayed reward. Utility theories for time mainly revolve around how individuals value their time and how they make decisions involving time. Here are some predominant theories:
> > > > > > > > > > > > > > > > > > >
> > > > > > > > > > > > > > > > > > > 1. **Exponential Discounting**: This theory assumes that people discount the future at a constant rate. The value of a future reward is reduced by a fixed percentage for each unit of time that passes.
> > > > > > > > > > > > > > > > > > >
> > > > > > > > > > > > > > > > > > > 2. **Hyperbolic Discounting**: Unlike exponential discounting, hyperbolic discounting assumes that people discount the future at a decreasing rate. The immediate future is heavily discounted, but the rate of discounting slows down for more distant future rewards.
> > > > > > > > > > > > > > > > > > >
> > > > > > > > > > > > > > > > > > > 3. **Quasi-Hyperbolic Discounting**: This model is a refinement of hyperbolic discounting and includes two parameters: one that reflects immediate discounting and another that captures the steady discounting of future time periods.
> > > > > > > > > > > > > > > > > > >
> > > > > > > > > > > > > > > > > > > 4. **Time Inconsistency Models**: These models highlight the inconsistencies in people's preferences over time, reflecting the idea that preferences today might differ from preferences in the future.
> > > > > > > > > > > > > > > > > > >
> > > > > > > > > > > > > > > > > > > 5. **Preference Learning Models**: Some newer approaches consider that individuals learn their preferences over time. The preferences aren't fixed but evolve with experiences and reflections.
> > > > > > > > > > > > > > > > > > >
> > > > > > > > > > > > > > > > > > > Models such as these are used in many fields, including economics, psychology, and sometimes in AI. They model human behavior in various contexts, like saving, investing, consuming, and other time-related decisions.

---

> > > > > > > > > > > > > > > > > > > > ### Comment · Reviewer_Qzq9 · 2023-08-18
> > > > > > > > > > > > > > > > > > > > **This is unfortunately irrelevant**
> > > > > > > > > > > > > > > > > > > >
> > > > > > > > > > > > > > > > > > > > This all is not relevant to the problem with the paper. I am familiar with submodular, supermodular, risk-seeking, risk-averse, convex, concave and other utilities.
> > > > > > > > > > > > > > > > > > > >
> > > > > > > > > > > > > > > > > > > > The problem is not linearity or non-linearity of the utility function. The problem is with it being bounded from below with respect to the running time. The authors discuss, as an example, a setting of hard time limit. According to the authors' approach, the utility at a beyond the hard time limit is 0. Which leads to the paradoxical situation in which the expected utility of an algorithm that does not return an answer within the hard time limit half of the time (or 99% of the time) can be higher than the expected utility of an algorithm which always returns an answer within the time limit.
> > > > > > > > > > > > > > > > > > > >
> > > > > > > > > > > > > > > > > > > > Discounting, sub/supermodularity, concavity and convexity are all good and I am aware of them, but the utility (or part of it) associtated with an infinite expense should be negative infinity, regardless of how slow the utility function approaches said negative infinity.
> > > > > > > > > > > > > > > > > > > >
> > > > > > > > > > > > > > > > > > > > Authors' own examples do not work with their theory, the authors unfortunately do not make these sanity checks, sorry.
> > > > > > > > > > > > > > > > > > > >
> > > > > > > > > > > > > > > > > > > > As a side note, **ALL UTILITY FUNCTIONS** you provided in section **Utility for money** increase with the argument. These are utility functions for money received, rather than money spent. This is not the same thing, and I would expect you, as the area chair, to check your statements before you post them, effectively taking a side in the discussion.

---

> > > > > > > > > > > > > > > > > > > > > ### Comment · Reviewer_Qzq9 · 2023-08-18
> > > > > > > > > > > > > > > > > > > > > **here is what your suggested utility functions do for negative arguments**
> > > > > > > > > > > > > > > > > > > > >
> > > > > > > > > > > > > > > > > > > > > $u(x) = x$ --- goes to negative infinity for $x \to -\infty$
> > > > > > > > > > > > > > > > > > > > >
> > > > > > > > > > > > > > > > > > > > > $u(x) = \log (x)$ --- same.
> > > > > > > > > > > > > > > > > > > > >
> > > > > > > > > > > > > > > > > > > > > $u(x) = \frac x {1 + x} --- **increases** to 1 as $x \to -\infty$.
> > > > > > > > > > > > > > > > > > > > >
> > > > > > > > > > > > > > > > > > > > > How does it help the discussion to bring as examples functions which either go to $-\infty$ (as they should and opposite to what the authors suggest) or **increase** with the expenses?

---

> > > > > > > > > > > > > > > > > > > > > ### Comment · Area_Chair_2iVp · 2023-08-18
> > > > > > > > > > > > > > > > > > > > >
> > > > > > > > > > > > > > > > > > > > > > According to the authors' approach, the utility at a beyond the hard time limit is 0. Which leads to the paradoxical situation in which the expected utility of an algorithm that does not return an answer within the hard time limit half of the time (or 99% of the time) can be higher than the expected utility of an algorithm which always returns an answer within the time limit.
> > > > > > > > > > > > > > > > > > > > >
> > > > > > > > > > > > > > > > > > > > > I am not sure I am understanding your argument and how it relates to Economic models. A flat utility after a certain cutoff seems quite reasonable. Many theories of discounting favor a 50/50 chance at getting 100 dollars tomorrow over a 100 percent chance of getting 100 dollars in 1000 years and would value of getting 100 dollars in 1000 years to be the same as getting 100 dollars in 1,000,000 years. I personally would rather have a 50/50 chance of getting 100 dollars tomorrow over a 100 percent chance of getting 100 dollars in 20 years.
> > > > > > > > > > > > > > > > > > > > >
> > > > > > > > > > > > > > > > > > > > > With respect to completion time, if I need to use the output of a job for a deadline this weekend, having the output next week or the week after have the same utility for me.

---

> > > > > > > > > > > > > > > > > > > > > > ### Comment · Reviewer_Qzq9 · 2023-08-18
> > > > > > > > > > > > > > > > > > > > > >
> > > > > > > > > > > > > > > > > > > > > > ## Submission under consideration
> > > > > > > > > > > > > > > > > > > > > >
> > > > > > > > > > > > > > > > > > > > > > We are discussing a submission which introduces an algorithm configuration approach based on utility which flats out at 0  on positive time infinity. An algorithm is chosen based on expected utility. If the utility flattens at 0 at positive time infinity, non-terminating algorithms can be chosen over terminating within any time limit.
> > > > > > > > > > > > > > > > > > > > > >
> > > > > > > > > > > > > > > > > > > > > > The utility has to go to negative infinity on positively infinite time for expected utility to be consistent.
> > > > > > > > > > > > > > > > > > > > > >
> > > > > > > > > > > > > > > > > > > > > > ## Economic models
> > > > > > > > > > > > > > > > > > > > > >
> > > > > > > > > > > > > > > > > > > > > > Economic models  discuss the **intrinsic utility** of 100 dollars now and at a future time. This intrinsic utility may itself decrease (for various modelling and practical reasons). This has nothing to do with the total utility of getting 100 dollars in 1000000, or 100 or 10 years, or 1 day, if you have to pay for the time. In other words, the intrinsic utility of 100 dollars over time is say,
> > > > > > > > > > > > > > > > > > > > > > $U_i(t) = 100\exp(-\alpha t)$, for some $\alpha$. The cost of obtaining that utility is still $C(t) = \beta t$, for some beta, or another infinitely increasing function of $t$. The total utility is $U = U_i(t) - C(t)$ and goes to negative utility. If to get 100 dollars you need to sustain yourself at 10 dollars per days for 10 years, it is -3550 dollars total utility.

---

> > > > > > > > > > > > > > > > > > > > > > > ### Comment · Reviewer_Qzq9 · 2023-08-18
> > > > > > > > > > > > > > > > > > > > > > >
> > > > > > > > > > > > > > > > > > > > > > > Algorithm configuration, in the form considered in the paper, relies on assumption that the intrinsic utility of the solution does not depend on the time at which the solution was obtained (for example, all algorithms always return an optimal solution if at all). Algorithms are chosen based on the expected cost to obtain that intrinsic utility.
> > > > > > > > > > > > > > > > > > > > > > >
> > > > > > > > > > > > > > > > > > > > > > > For expected cost to comply with the utility theory, in economics, physics, psychology, or astronomy, the cost at positive resource (time) infinity must be positive infinity, that is the total utility must be negative utility at positively infinite investment.

---

> > > > > > > > > > > > > > > > > > > > > > ### Comment · Area_Chair_2iVp · 2023-08-18
> > > > > > > > > > > > > > > > > > > > > >
> > > > > > > > > > > > > > > > > > > > > > Here’s a very concrete example. Suppose you would like to buy a house in my town, and there are several houses for sale, but some are much prettier than others and the pretty ones are likely to be sold sooner. Suppose you are running a bitcoin mining algorithm, that when it completes will give you enough money to buy any of the houses. If it never completes, you can continue to live with your parents. :-) so you might prefer a 50-50 chance of the algorithm completing very soon and getting a really nice house or it completing at a much later date when it’s too late to buy a house over a 100% chance of it completing later, when you can only buy an ugly house.

---

> > > > > > > > > > > > > > > > > > > > > > > ### Comment · Reviewer_Qzq9 · 2023-08-18
> > > > > > > > > > > > > > > > > > > > > > > **intrinsic vs. total**
> > > > > > > > > > > > > > > > > > > > > > >
> > > > > > > > > > > > > > > > > > > > > > > Mind you, mining bitcoin is expensive. The total utility of running algorithm infinitely long is infinitely negative.

---

> > > > > > > > > > > > > > > > > > > > > > > ### Comment · Reviewer_Sepd · 2023-08-18
> > > > > > > > > > > > > > > > > > > > > > >
> > > > > > > > > > > > > > > > > > > > > > > Other reviewer here -- sorry, I do not understand the example in the context of algorithm configuration.
> > > > > > > > > > > > > > > > > > > > > > >
> > > > > > > > > > > > > > > > > > > > > > > Question for Reviewer Qzq9: (I apologize in advance for my ignorance; I work on AC, not utility functions, although I am greatly interested by the discussion to learn more). You said "... Which leads to the paradoxical situation in which the expected utility of an algorithm that does not return an answer within the hard time limit half of the time (or 99% of the time) can be higher than the expected utility of an algorithm which always returns an answer within the time limit."
> > > > > > > > > > > > > > > > > > > > > > >
> > > > > > > > > > > > > > > > > > > > > > > Consider two configurations for a SAT solver and a training dataset of 100 instances. Assume I want to minimize the runtime, as the solver is exact.
> > > > > > > > > > > > > > > > > > > > > > >
> > > > > > > > > > > > > > > > > > > > > > > 1. Configuration A times out one one instance, but solves the other 99 in one second.
> > > > > > > > > > > > > > > > > > > > > > > 2. Configuration B solves all 100 instances in the timeout - 1 second.
> > > > > > > > > > > > > > > > > > > > > > >
> > > > > > > > > > > > > > > > > > > > > > > Clearly there are some users who would prefer configuration A and some who would prefer B. Current AC approaches can be configured to prefer either situation (without any utility functions). Does this scenario counter the "paradoxical" situation you point out or have I misunderstood something?

---

> > > > > > > > > > > > > > > > > > > > > > > > ### Comment · Reviewer_Qzq9 · 2023-08-18
> > > > > > > > > > > > > > > > > > > > > > > > **what does "times out" mean?**
> > > > > > > > > > > > > > > > > > > > > > > >
> > > > > > > > > > > > > > > > > > > > > > > > The approach proposed by the authors implicitly assumes that an algorithm that times out may run _forever_, because the utility is at least 0 anyway. This is not the situation you are describing. I assume that when you say "times out" you mean that it is forcibly stopped and the solution is discarded. The intrinsic utility of the discarded solution is 0.
> > > > > > > > > > > > > > > > > > > > > > > >
> > > > > > > > > > > > > > > > > > > > > > > > Let us assume that the intrinsic utility of the accepted solution is 1 (or any other constant). Then, the total expected utility of the first algorithm is 0.99*(1 - 1second * cost_of_1second) + 0.01*(0 - timeout * cost_of_1second). The total expected utility of the second algorithm is 1 - (timeout - 1second) *cost_of_1second. Depending on the cost, you choose one or the other.
> > > > > > > > > > > > > > > > > > > > > > > >
> > > > > > > > > > > > > > > > > > > > > > > > However, for algorithm configuration, this requires dealing with long tails of execution times. The submission proposes a magic trick of passing the time cost through a sigmoid-like transformation to avoid dealing with the long tails. This is not going to work in practice, you will either have long tails anyway, or will select wrong algorithms.

---

> > > > > > > > > > > > > > > > > > > > > > > > > ### Comment · Reviewer_Sepd · 2023-08-18
> > > > > > > > > > > > > > > > > > > > > > > > >
> > > > > > > > > > > > > > > > > > > > > > > > > Thanks for the explanation. In AC, we always have a cutoff/timeout for the algorithms under consideration. Anything else makes no sense.

---

> > > > > > > > > > > > > > > > > > > > > > > > > > ### Comment · Area_Chair_2iVp · 2023-08-18
> > > > > > > > > > > > > > > > > > > > > > > > > > **End this thread**
> > > > > > > > > > > > > > > > > > > > > > > > > >
> > > > > > > > > > > > > > > > > > > > > > > > > > Let’s end this thread and try other means to resolve.

---

> > > > > > > > > > > > > > > > ### Comment · Reviewer_Qzq9 · 2023-08-18
> > > > > > > > > > > > > > > >
> > > > > > > > > > > > > > > > Dear area chair, would you like me to continue the discussion or stop here?

---

> ### Comment · Reviewer_Qzq9 · 2023-08-20
> **Graham et al 2022**
>
> I've read the cited paper. It is not my assignment here to review it (it was already accepted elsehwere), however, it has certain serious problems which I would like to point here, because this submission cites this paper extensively.

---

> > ### Comment · Reviewer_Qzq9 · 2023-08-20
> > **Graham et al (sent the comment accidentially too early)**
> >
> > First, the paper is ignorant about the probability and statistics. On page 1, second column, third paragraph, the paper states as an obvious fact that a distribution on an infinite domain does not have a mean. While there are examples of distributions that do not have a mean (e.g. Cauchy) many distributions, both discrete and continuous, defined on infinite domains, do have means. Possion, Geometric, Exponential, Normal, etc.
> >
> > Second, the paper has a section on estimating the mean of a distribution based on right-censored samples. This is a known problem in statistics, there are nice methods for that.
> >
> > Third, the main confusion in the paper is an implicit assumption that an utility (or the expected utility) cannot be negative. The paper introduces the intrinsic utility and the cost separately, and then carefully redefines the net utility so that it is some decreasing function if positive, and 0 everywhere else.
> >
> > It is totally normal for an utility (of an instance or of an expected utility) to be negative. And the utility-based decision rule is simple --- don't take an action if the utility is negative. Written in the books for last several decades. Most complicated theoretical exercises in that paper (Graham et al) is that the authors somehow got stuck on keeping the utility non-negative. It does not have to be. It can be negative, and if the expected utility is negative, you lose more than you gain, on average, and should not take that action.
> >
> >
> > The submission uses some of the results of Graham's paper, including the wrong ones.

---

> > > ### Comment · Area_Chair_2iVp · 2023-08-20
> > >
> > > The authors should not feel responsible for defending the previously published paper of Graham et al (2022)

---

> > > > ### Comment · Reviewer_Qzq9 · 2023-08-20
> > > > **Responsibility**
> > > >
> > > > Of course they should not feel responsible for defensibg a previously paper. They however should feel responsible for reading critically the paper they base their work and their rebuttal upon. The authors cite the paper multiple time and appeal to the paper in their rebuttal as though it is the undisputed truth. The paper appears to be problematic.

---

### Decision · Program_Chairs · 2023-09-21

**Decision:**

Accept (poster)

**Comment:**

Discounting the score of Reviewer Qzq9 (whose remarks were largely inconsistent with prior established work across fields), the other reviewers all liked this work and felt it was a valuable contribution to the field of algorithmic configuration.